# *Mir-132/212* is required for maturation of binocular matching of orientation preference and depth perception

Raffaele Mazziotti[1], Laura Baroncelli[2], Nicholas Ceglia[3,4], Gabriele Chelini[1], Grazia Della Sala[1], Christophe Magnan[3,4], Debora Napoli[5], Elena Putignano[2], Davide Silingardi[1], Jonida Tola[2], Paola Tognini[5,6,7], J. Simon C. Arthur[8], Pierre Baldi[3,4,6,7] & Tommaso Pizzorusso[1,2,5]

MicroRNAs (miRNAs) are known to mediate post-transcriptional gene regulation, but their role in postnatal brain development is still poorly explored. We show that the expression of many miRNAs is dramatically regulated during functional maturation of the mouse visual cortex with miR-132/212 family being one of the top upregulated miRNAs. Age-down-regulated transcripts are significantly enriched in miR-132/miR-212 putative targets and in genes upregulated in *miR-132/212* null mice. At a functional level, *miR-132/212* deletion affects development of receptive fields of cortical neurons determining a specific impairment of binocular matching of orientation preference, but leaving orientation and direction selectivity unaltered. This deficit is associated with reduced depth perception in the visual cliff test. Deletion of *miR-132/212* from forebrain excitatory neurons replicates the binocular matching deficits. Thus, miR-132/212 family shapes the age-dependent transcriptome of the visual cortex during a specific developmental window resulting in maturation of binocular cortical cells and depth perception.

[1] Department of Neuroscience, Psychology, Drug Research and Child Health NEUROFARBA University of Florence, Area San Salvi—Pad. 26, 50135 Florence, Italy. [2] Institute of Neuroscience, National Research Council, Via Moruzzi, 1 56124 Pisa, Italy. [3] Department of Computer Science, University of California, Irvine, Irvine, California 92697, USA. [4] Institute for Genomics and Bioinformatics, University of California, Irvine, Irvine, California 92697, USA. [5] BIO@SNS lab, Scuola Normale Superiore via Moruzzi, 1 56124 Pisa, Italy. [6] Department of Biological Chemistry, University of California, Irvine, Irvine, California 92697, USA. [7] Center for Epigenetics and Metabolism, University of California, Irvine, Irvine, California 92697, USA. [8] Division of Cell Signalling and Immunology, School of Life Sciences, University of Dundee, Dundee, UK. Correspondence and requests for materials should be addressed to T.P. (email: tommaso.pizzorusso@in.cnr.it).

MicroRNAs (miRNAs) are an endogenous class of small non-coding RNAs that regulate gene expression by causing degradation or translational repression of their target messenger RNAs (mRNAs)[1]. In the last decades, miRNAs have been involved in the physiology of all tissues through the control of a wide variety of signalling pathways. At cerebral level, miRNAs have been mainly studied during early phases of development[2] or in models of neuropathological disease[3] and neural plasticity[4–7]. Recent evidence has demonstrated that visual experience is able to influence cortical levels of specific miRNAs[8,9] and that tight regulation of miR-132 levels is required for ocular dominance (OD) plasticity during a sensitive period of development[8–11]. However, neither miR-132 nor other miRNAs have ever been investigated in the context of physiological maturation of cortical functions. Studies performed in rodents have shown that visual cortical neurons display functional properties, such as orientation and direction selectivity, emerging during early windows of development and relatively insensitive to visual experience[12–15], whereas other properties mature later during development and are more sensitive to the visual input[13,16–18]. A remarkable example is the binocular matching of orientation preference that develops during the critical period (CP) for OD plasticity and is strongly dependent on visual experience[19–23]. Indeed, before CP opening neuronal visual responses show a different preferred orientation between the two eyes that gradually matches during the CP. Eventually, adult cortical neurons show similar orientation preference for the two eyes, albeit some misaligned cells can persist[19,24,25]. We reasoned that miRNAs undergoing developmentally regulated expression during this period of functional maturation are likely candidates as key molecular mediators of cortical development. Thus, we analysed miRNAs regulated during visual cortical development by performing an RNA sequencing analysis of miRNAs expressed in the visual cortex immediately before eye opening (P10) and when molecular mechanisms underlying development of mouse visual function are fully engaged (P28; refs 12,14,15,18,19). To investigate miRNA targets, mRNAs from the same samples were also analysed. We found that many miRNAs undergo a strong expression change during this period of development. In particular, miR-132 family, represented by miR-132-3p and miR-212-5p, was one of the top three upregulated miRNAs with a corresponding significant downregulation of its putative target mRNAs. Therefore, we decided to study age-dependent transcriptomic regulation and the development of visual cortical receptive fields (RF) in mice with deletion of the genomic locus coding for both miR-132 and miR-212 (ref. 26). These mice showed an upregulation of miR-132/212 targets that are normally downregulated with age, and displayed a selective impairment in the binocular matching of orientation preference of cortical units similar to that observed in monocularly deprived (MD) wild-type (wt) mice. Strikingly, the same effect was present when *miR-132/212* was selectively deleted in the excitatory neurons of the forebrain. From a behavioural point of view, *miR-132/212* null animals exhibited impaired depth perception in the visual cliff test. Finally, the deficit in binocular matching and stereoscopic vision was still persistent in adult *miR-132/212* null mice suggesting that miR-132/212 is necessary to orchestrate the adaptive refinement of visual circuits during a specific sensitive period of development.

## Results

### MiR-132 affects age-dependent visual cortical transcriptome.

To investigate miRNA expression during postnatal development we performed RNA sequencing of small RNAs extracted from the visual cortex of P10 and P28 mice. A total of 176,228,828 raw reads were generated. On average 94.5% (s.d. = 0.64) of the reads could be aligned. 2,164 isomiRs (Supplementary Data 1) and 299 precursors (Supplementary Data 1 and Fig. 1a) were found to have age-regulated expression. Mir-29a, miR-219, miR-338 and miR-132 were the miRNAs undergoing the strongest upregulation during development, a result confirmed by reverse transcription PCR (Supplementary Fig. 1) and in agreement with previous data[8], whereas miR-298, miR-149 and miR-331 were the top downregulated miRNAs. The corresponding miRNA families were also the strongest regulated families (Supplementary Data 1). Among the members of the miR-132 family, miR-132-3p and miR-212-5p were the only miRNAs represented at high levels.

Then, we investigated the impact of developmental regulation of miR-132-3p and miR-212-5p on gene expression by analysing the transcriptome in the same samples used for the small RNA sequencing. 4,339 genes were significantly upregulated and 5,429 genes were downregulated in the P28 visual cortex with respect to the P10 cortex (Supplementary Data 2). Kyoto Encyclopedia of Genes and Genomes (KEGG) pathway analysis (Fig. 1b,c) revealed that some pathways were in common between upregulated (Supplementary Data 2) and downregulated (Supplementary Data 2) genes, whereas other pathways were significantly enriched only in one of these groups. Many pathways previously involved in cortical development and plasticity were affected[27–31] strengthening the predictive validity of our analysis. Indeed, MAPK signalling, neurotrophin signalling, glutamatergic synapse, neuroactive ligand-receptor interaction, insulin signalling pathways were significantly enriched in both upregulated and downregulated genes; regulation of actin cytoskeleton, circadian rhythm—mammal and chemokine signalling pathways were present only in upregulated genes; axon guidance, RNA transport, gap junction, long-term potentiation, long-term depression and mTOR signalling pathways were present only in the downregulated genes. Many of the developmentally regulated genes were predicted targets of multiple miRNAs (Supplementary Data 2). Importantly, there was a highly significant and specific overlap between the predicted targets of miR-132-3p and the genes downregulated with age (Supplementary Data 2, 181 genes, odds ratio 2.10; Fisher exact test $P < 0.0001$). MiR-212-5p targets were also significantly enriched in age-downregulated genes albeit with a minor odds ratio than miR-132-3p (Supplementary Data 2, 132 genes, odds ratio = 1.74; Fisher exact test $P < 0.0001$). This result is in agreement with the hypothesis that miR-132/212 age-regulated increase contributes to repress the expression of a significant number of genes during visual cortical development.

To independently test this hypothesis, we performed RNA sequencing on P28 visual cortical samples obtained from mice with germ-line deletion of the *miR-132/212* locus. 1698 genes were differentially expressed between mutant and wt mice (Supplementary Data 3). Intriguingly, KEGG pathway analysis revealed that 53 out of the 61 KEGG categories enriched with genes upregulated in *miR-132/212* null mice were also present in the KEGG categories downregulated during normal development, suggesting that a substantial part of the rearrangement in molecular pathways occurring during normal development is altered in *miR-132/212* mutants. Moreover, a significant enrichment in miR-132-3p targets was present in the genes upregulated in the *miR-132/212* mutant cortex (54 genes, odds ratio 5.07; Fisher exact test $P < 0.0001$, Supplementary Data 3), whereas the enrichment in miR-212-5p targets was not significant (14 genes, odds ratio 1.50; Fisher exact test $P = 0.13$, Supplementary Data 3). Importantly, there was an overlap (39 genes) between the miR-132-3p targets and genes that were both downregulated by age and upregulated by *miR-132/212* deletion (Fig. 1d). This gene set

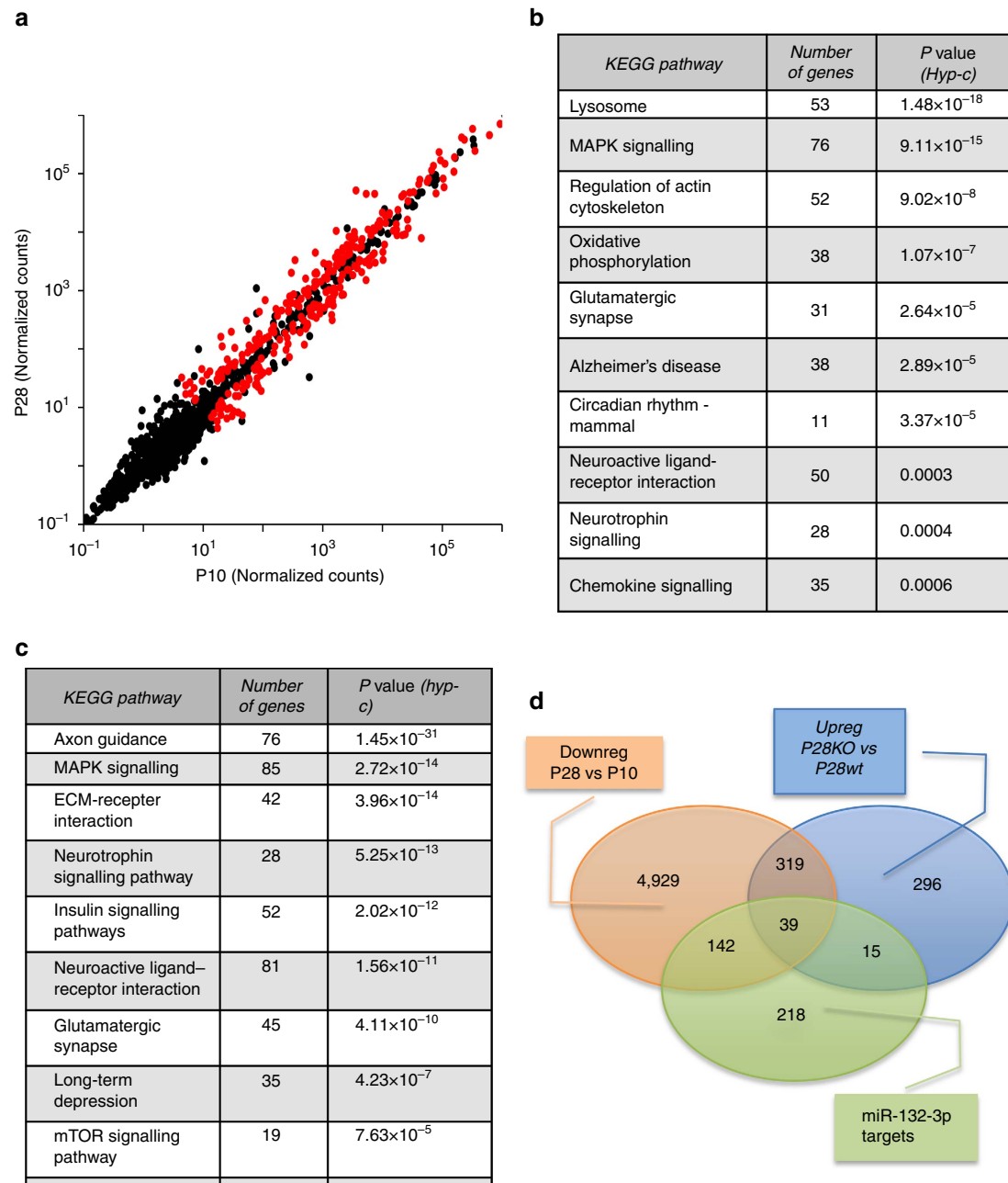

**Figure 1 | RNA and small RNA sequencing reveal a role of miR-132/212 in shaping the age-regulated transcriptome.** (**a**) Scatter plot of miRNAs regulated between P10 and P28 in the mouse visual cortex ($N = 4$ mice for each age). Red dots represent significantly regulated miRNAs. (**b**) KEGG pathways significantly upregulated with age. (**c**) KEGG pathways significantly downregulated with age. (**d**) Venn diagram showing the intersection between age-downregulated genes, genes upregulated in P28 *miR-132/212* null mice ($N = 4$ mice for each genotype) and putative miR-132-3p target genes.

included genes important for brain development such as *MeCP2, Sox5, Sox11* and *Pten* (Supplementary Data 4). Thus, in the absence of miR-132 family the developmental downregulation of a significant number of miR-132-3p targets does not occur, confirming the importance of miR-132-3p developmental regulation in defining the transcriptomic changes occurring between P10 and P28 in the visual cortex.

**Mir-132/212 null mice show normal monocular RF properties.** We investigated whether the lack of *miR-132/212* cluster was able to influence functional development of the visual cortex. We first analysed monocular tuning properties in sorted units recorded by

multisite silicon electrode tetrodes inserted at depths sampling from layers III to V in binocular visual cortex of P27-28 wt and null mice. Orientation and direction selectivity was measured on responses to drifting sinusoidal gratings calculating three different parameters: orientation selectivity index (OSI), orientation tuning width and direction selectivity index (DSI)[15,32]. Each index was separately computed for the contralateral and ipsilateral eye responses. No difference between wt and null mice was present for all these indexes (One-way analysis of variance (ANOVA), $P = 0.159$, $P = 0.595$ and $P = 0.262$ respectively; Fig. 2a–e) indicating that the maturation of these properties does not require *miR-132/212*.

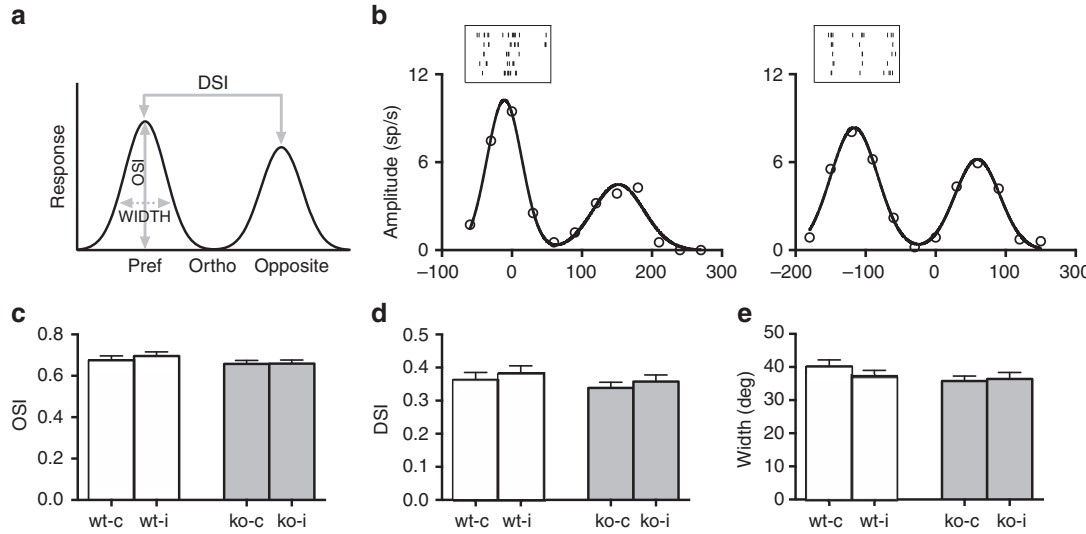

**Figure 2 | *MiR-132/212* deletion does not affect monocular properties of cortical receptive fields.** (**a**) Schematic representation of metrics used for characterization of single-unit responses. (**b**) Orientation tuning curves of a visually responsive unit for wt (left, $N = 6$ mice, $n = 132$ cells) and *miR-132/212* mutant (right, $N = 6$ mice, $n = 186$ cells) mice. The black lines are fitted curves of the response magnitudes (open circles). Spike rasters of representative unit phase-locked response to preferred drifting sine gratings are also depicted. No difference in orientation selectivity (**c**), orientation tuning width (**d**) and direction selectivity (**e**) is observed in contralateral and ipsilateral responses of mutant mice compared to wt animals. Error bars represent s.e.m.

***Mir-132/212* null mice show impaired binocular matching**. Previous data showed that after orientation selectivity completes its developmental trajectory, there is a process of binocular matching of preferred orientation[19]. To further analyse the role of miR-132/212 in visual cortical development, we assessed binocular matching of orientation preference in P27-28 null and wt mice. We found that mutant mice had a significantly worse binocular matching of orientation preference with respect to wt age-matched littermates (Two-way ANOVA, effect of genotype $P < 0.001$; *post hoc* Holm-Sidak test, $P < 0.01$; Fig. 3a,b), suggesting that the lack of miR-132 family specifically disrupts this late developing property of visual cortical neurons while keeping monocular tuning properties intact. Intriguingly, the defective binocular matching of orientation preference present in P27-28 *miR-132/212* null mice persists into adulthood. Indeed, we measured binocular matching in adult P60 mutant and wt mice and we found that also at this age binocular matching was dramatically impaired in *miR-132/212* null mice (*post hoc* Holm-Sidak test, $P < 0.001$; Fig. 3a), despite a normal orientation selectivity of visual cortical neurons (Two-way ANOVA, effect of genotype $P = 0.133$; Supplementary Fig. 2A).

Single-unit data pooled from many animals can be biased towards those individual animals in which the largest number of units were studied. Therefore, we next analysed the results by case. Even with this analysis, binocular matching resulted to be strongly impaired in *miR-132/212* null mice both at P27-28 and P60 (Two-way ANOVA, effect of genotype $P < 0.001$; *post hoc* Holm-Sidak test, wt versus KO: P27-28 $P < 0.01$, P60 $P < 0.001$; Supplementary Fig. 2B).

Visual cortical neurons can be classified into simple and complex cells based on their response properties[32,33]. Since it has been recently reported that during physiological development simple cells fulfil the process of binocular matching of orientation preference before complex cells[19–23], we studied the degree of binocular matching in simple and complex cells of null mice. Simple cells of null mice only showed a statistically nonsignificant trend for binocular mismatching with respect to wt animals (*t*-test, $P = 0.07$; Fig. 3c), while the binocular matching of complex cells was significantly disrupted (*t*-test, $P < 0.01$; Fig. 3d).

Consistently with the results reported for the whole-cell population, monocular orientation tuning properties of both simple and complex cells were normal (One-way ANOVA $P = 0.624$ and $P = 0.524$, respectively; Fig. 3e,f). These results are in line with the hypothesis that miR-132 family is particularly important for the late development of binocular neuronal properties.

Since narrow-spiking units are known to be poorly orientation selective[32], an abnormally high presence of narrow-spiking units in *miR-132/212* null mice could contribute to the low binocular matching of orientation preference observed in mutant mice. To test this possibility we classified neuronal units into two different classes, narrow spiking and broad spiking (Fig. 3g), on the basis of their spike waveform. This waveform signature is used to distinguish putative excitatory and fast spiking inhibitory neurons[32]. We found that 5.4% of *miR-132/212* mutant units belonged to the narrow-spiking class. This percentage was not significantly different from that detected in the cortex of wt animals (8.4%; Fisher's exact test $P = 0.36$) and in line with previous studies on mice of the same age[12–15]. These data show that the deletion of *miR-132/212* locus does not impinge on the development of narrow-spiking units suggesting that the disruption of binocular matching in null mice was not due to an alteration of narrow-spiking inhibitory neurons. Consistently, the analysis of binocular matching of orientation preference exclusively in broad-spiking cells showed a significant disruption of this property in *miR-132/212* mutants (*t*-test, $P < 0.01$; Fig. 3h) with no change in orientation selectivity (One-way ANOVA $P = 0.165$; Fig. 3i). These results also suggest that miR-132 could be mainly involved in the maturation of response properties of excitatory cortical neurons.

**No OD plasticity and binocular matching in *miR-132/212* mice**. It has been reported that binocular matching of orientation preference is an experience-dependent process[19] sharing molecular regulatory mechanisms with OD plasticity[23]. Thus, we decided to investigate whether OD plasticity was blocked in *miR-132/212* null animals and whether the effects of MD on

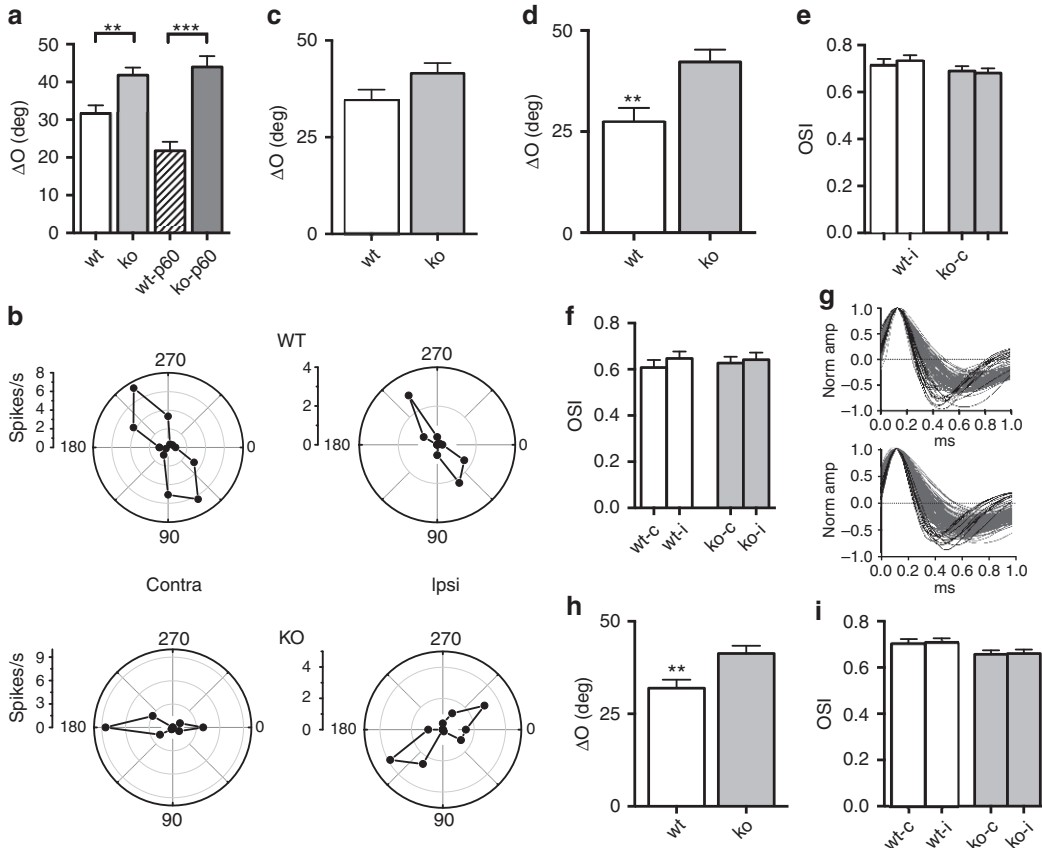

**Figure 3 | Binocular matching of orientation preference is impaired in *miR-132/212* null mice.** (**a**) Mutant mice (ko, $N = 6$ mice, $n = 186$ cells; ko-P60, $N = 5$ mice, $n = 85$ cells) have a significantly higher difference in preferred orientation ($\Delta$O) with respect to wt animals both at P30 and P60 (wt, $N = 6$ mice, $n = 132$ cells; wt-P60, $N = 5$ mice, $n = 59$ cells). (**b**) Representative polar plots of orientation-tuned responses from contralateral and ipsilateral eye are depicted. (**c**) $\Delta$O of simple cells is comparable in wt and mutant mice. (**d**) $\Delta$O of complex cells is significantly impaired in mutant mice with respect to wt animals. Normal orientation selectivity is detected in mutant simple (**e**) and complex cells (**f**). (**g**) Waveforms of visually responsive cells clustered into narrow spiking (black) and broad spiking (grey) for wt (top) and mutant mice (bottom). The average spike waveform for each unit is plotted aligned to its maximum and normalized by trough depth. (**h**) $\Delta$O of broad-spiking units is higher in mutant mice with respect to wt animals. (**i**) No change in orientation selectivity is observed in broad-spiking units. Asterisks indicate statistical significance. Error bars represent s.e.m.

binocular matching could be occluded by *miR-132/212* deletion. Previous work had already shown that miR-132 is necessary for OD plasticity[9], however a temporally restricted block of miR-132 availability using a miR-132 sponge was adopted. Therefore, we first controlled whether OD plasticity was also blocked by our genetic deletion of *miR-132/212*. We recorded visual evoked potentials (VEPs) and single units in wt and null mice MD for 3 days from P24-25. VEP recordings showed that non-deprived null mice have a normal OD ratio compared to age-matched wt littermates (Two-way ANOVA on rank transformed data, *post hoc* Holm-Sidak test $P = 0.566$), and that three days of MD were not able to induce OD shift in mutant mice (Two-way ANOVA on rank transformed data, *post hoc* Holm-Sidak test $P = 0.958$; Fig. 4a). MD in wt animals led to a significant decrease in the C/I VEP ratio (Two-way ANOVA on rank transformed data, genotype $\times$ condition interaction $P = 0.01$; *post hoc* Holm-Sidak test $P < 0.001$).

Single-unit analysis of the ocular dominance index (ODI) confirmed the lack of OD plasticity in null mice. ODI of null mice was not significantly different from that of wt animals (Two-way ANOVA, *post hoc* Holm-Sidak test $P = 0.210$) and from that of MD null mice (Two-way ANOVA, *post hoc* Holm-Sidak test $P = 0.652$), whereas MD wt mice displayed the typical OD shift towards the open eye (Two-way ANOVA, genotype $\times$ condition

interaction $P < 0.05$; *post hoc* Holm-Sidak test $P < 0.01$; Fig. 4b). Contralateral bias index (CBI) expressing the strength of contralaterally driven responses for each animal confirmed the results of unit-based analysis (Two-way ANOVA, genotype $\times$ condition interaction $P < 0.05$; *post hoc* Holm-Sidak test: wt versus ko $P = 0.96$, ko versus ko-md $P = 0.97$, wt versus wt-md $P < 0.05$; Supplementary Fig. 3A).

Then, we analysed the influence of visual experience on binocular matching level in P27-28 *miR-132/212* null and wt mice subjected to a 3-days MD. We found that the effect of MD on binocular matching was occluded by the deletion of *miR-132/212*: while wt-md animals displayed a significant impairment of binocular matching with respect to non-deprived mice (Two-way ANOVA, genotype $\times$ condition interaction $P < 0.05$, *post hoc* Holm-Sidak test $P < 0.001$), the closure of one eye did not further deteriorate the mismatch of orientation preference observed in *miR-132/212* null animals (*post hoc* Holm-Sidak test $P = 0.767$; Fig. 4c). The same conclusion emerged from the analysis of binocular matching by case (Two-way ANOVA, genotype $\times$ condition interaction $P < 0.01$; *post hoc* Holm-Sidak test: wt versus wt-md $P < 0.01$, ko versus ko-md $P = 0.795$, Supplementary Fig. 3B). These results demonstrate that *miR-132/212* is necessary for experience-dependent development of binocular processes in the visual cortex.

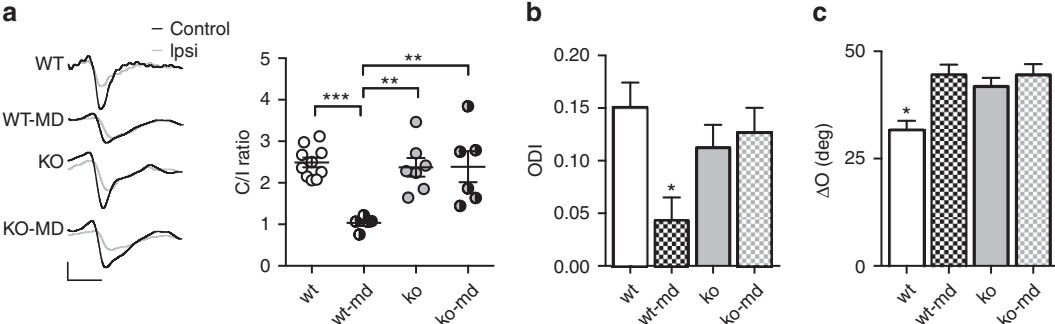

**Figure 4 | *MiR-132/212* deletion blocks OD plasticity and occludes monocular deprivation effects on binocular matching of orientation preference.**
(**a**) Contralateral to ipsilateral eye (C/I) VEP ratio in wt ($N = 10$ mice), MD wt (wt-md, $N = 5$ mice), *miR-132/212* null (ko, $N = 7$ mice) and MD *miR-132/212* null animals (ko-md, $N = 6$ mice). Circles represent individual C/I ratio for each animal. Typical VEP responses to the stimulation of either contralateral (black) or ipsilateral (grey) eye are reported (left). Calibration bars: 100 μV, 100 ms. (**b**) ODI values for wt ($N = 6$ mice, $n = 132$ cells), wt-md ($N = 6$, $n = 117$ cells), ko ($N = 6$ mice, $n = 186$ cells) and ko-md ($N = 5$; $n = 119$ cells) animals. Histograms represent the average ODI ± s.e.m. for each experimental group. (**c**) MD increases ΔO in wt animals, while no difference is present between ko and ko-md mice. Asterisks indicate statistical significance. Error bars represent s.e.m.

**Depth perception impairment in *miR-132/212* null mice**. We then asked whether the disruption of binocular matching for orientation preference caused by *miR-132/212* deletion in the visual cortex could affect animals' perception abilities. We first performed an electrophysiological assessment of visual acuity (VA) in P27-28 *miR-132/212* null mice[34]. No VA difference was present between mutant and wt animals (*t*-test, $P = 0.781$; Supplementary Fig. 4), confirming that molecular mechanisms underlying experience-dependent regulation of binocularity of cortical neurons can be dissociated from those involved in VA maturation and plasticity[18,31,35–37].

We then focused on depth perception, given the importance of binocular cues for this visual function. We employed the visual cliff task to explore the effects of *miR-132/212* deletion on stereoscopic visual abilities. This test exploits the spontaneous tendency of rodents to avoid the deep side of a visual cliff arena (Fig. 5a). The construct validity of the visual cliff exploration test was first assessed investigating the behaviour of mice in monocular condition (that is, subjected to the closure of one eye obtained through eyelid suture). As previously reported[38], animals in binocular condition (bin) spent a longer period of time on the shallow side of the arena, while monocular condition (mon) led to a significantly lower preference for the shallow side (*t*-test, $P < 0.01$; Fig. 5b). Total distance moved and velocity were comparable between the two groups (*t*-test, $P = 0.123$ and $P = 0.142$ respectively, Supplementary Fig. 5A,B). These results confirmed that binocular vision is required for discrimination between the deep and the shallow side. To exclude the contribution of non-visual cues to the preferential exploration of the shallow side, we analysed the behaviour of binocular wt animals in the same visual cliff arena with the two visual stimuli placed at the same height immediately below the glass plates. In this condition, the differential exploration of the two sides of the arena shown by binocular wt mice in the visually cued version was completely eliminated (*t*-test, $P < 0.01$). Indeed, wt mice equally explored the two sides of the arena (one-sample *t*-test versus 50%; wt $P = 0.514$, Fig. 5c). The same was true for *miR-132/212* null animals (one-sample *t*-test versus 50%; ko $P = 0.622$, Fig. 5c). These data demonstrate the tight relevance of visual cues in visual cliff test. No difference was present between wt and mutant mice in the time spent in the centre of the arena (*t*-test, $P = 0.663$, Fig. 5d) indicating similar levels of anxiety-like behavior in the two genotypes.

Next, we evaluated depth perception abilities of *miR-132/212* null mice. We found that *miR-132/212* mutant mice showed a significantly reduced preference for the shallow side with respect to wt mice (Kruskal–Wallis One-way ANOVA on ranks, *post hoc* Dunn's method, $P < 0.05$; Fig. 5e, Supplementary Fig. 5C and Supplementary Movies 1 and 2). Interestingly, the impairment in stereoscopic abilities detected in *miR-132/212* null mice was reminiscent of that observed in animals subjected to a 3-day MD and tested after the restoration of binocular vision (that is, 2 h after the reopening of the deprived eye, wt-md; Kruskal–Wallis One-way ANOVA on ranks, *post hoc* Dunn's method, Fig. 5e, Supplementary Fig. 5C). To rule out the possibility that the significant difference in visual capacities reflect changes in the ability to cope with stress in challenging task conditions, we analysed general activity and anxiety-related behaviour of wt and *miR-132/212* mutant mice in the visual cliff arena. We found that total activity levels of animals were not affected by the deletion of *miR-132/212* (distance moved: *t*-test, $P = 0.285$; velocity: *t*-test, $P = 0.303$; Fig. 5f,g). Moreover, the time spent by *miR-132/212* mutant mice in the central portion of the apparatus was not different from that recorded for wt animals (*t*-test, $P = 0.126$, Supplementary Fig. 5D), excluding the hypothesis that a combination of abnormal anxiety and activity levels might be related to their altered performance in the visual cliff arena.

In tight accordance with binocular matching results, depth perception impairment in *miR-132/212* null mice persisted in adulthood: stereoscopic abilities of mutants at P60, indeed, appeared markedly altered with respect to age-matched wt animals (*t*-test, $P < 0.05$; Fig. 5h, Supplementary Fig. 5E and Supplementary Movies 3 and 4). Time spent in the central part of the arena, total activity and velocity of animals were comparable between genotypes also in this case (*t*-test, $P = 0.767$, $P = 0.431$ and $P = 0.349$, respectively; Supplementary Fig. 5F, Fig. 5i,j). These results demonstrate that disruption of the binocular orientation tuning properties of neurons in the primary visual cortex is associated with behavioural deficits in depth perception.

**Impaired binocular matching in Emx1:Cre-miR-132/212$^{-/-}$ mice.** Previous studies showed that miR-132 family is preferentially expressed by excitatory cells[39]. Moreover, we found that *miR-132/212* deletion mainly affected the response properties of broad-spiking cells (Fig. 3e) and morphological analysis of *miR-132/212* null mice crossed with mice expressing green fluorescent protein in layer V pyramidal neurons showed a small but significant reduction spine density with respect to wt littermates (*t*-test, $P < 0.05$ Fig. 6a,b and Supplementary Fig. 6), suggesting that

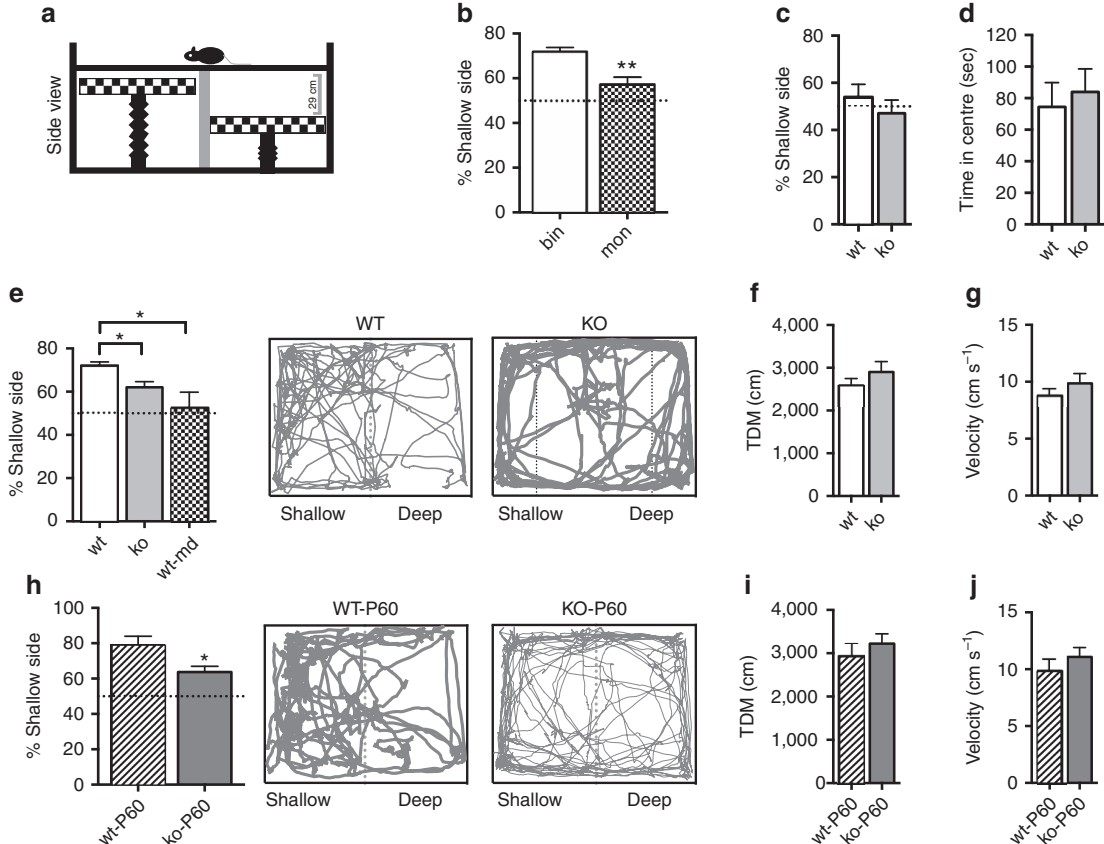

**Figure 5 | Impaired depth perception in *miR-132/212* null mice.** (**a**) Schematic diagram and components of the visual cliff. (**b**) Binocular (bin, $N = 10$) wt animals spend a longer period of time on the shallow side of the arena with respect to monocular mice (mon, $N = 8$). (**c**) wt ($N = 7$) and ko ($N = 9$) mice equally explored the two sides of the arena when the two visual stimuli were placed at the same height immediately below the glass plates (one-sample *t*-test versus 50%; wt $P = 0.514$, ko $P = 0.622$). (**d**) No difference was detected between wt and ko animals in the time spent in the centre of the arena with no visual depth cues (*t*-test, $P = 0.663$). (**e**) A significant preference for the shallow side of the visual cliff is detected for wt ($N = 10$), while no preference is observed in *miR-132/212* null (ko, $N = 9$) and MD wt mice (wt-md, $N = 7$). Representative examples of movement path for a wt (left) and a ko mouse (right) are reported. (**f**), (**g**) Total activity levels (TDM and VEL) of animals are not affected by the deletion of *miR-132/212* gene. (**h**) Average percentage of time spent in the shallow side of the arena for wt (wt-P60, $N = 7$) and *miR-132/212* null animals at P60 (ko-P60, $N = 7$). Representative examples of movement path for a wt-P60 (left) and a ko-P60 mouse (right) are reported. Total distance moved (**i**) and velocity (**j**) of wt-P60 and ko-P60 animals. Asterisks indicate statistical significance. Error bars represent s.e.m.

excitatory cells could be a specific cellular target of miR-132 family. To directly investigate this possibility, we generated a novel mouse model (Emx1:Cre-miR-132/212$^{-/-}$) in which floxed *miR-132/212* alleles are specifically deleted in forebrain glutamatergic neurons and in some glial cells by using the Emx1 promoter to drive Cre-recombinase expression[40]. The electrophysiological characterization of these mutants at P27-28 revealed no differences in monocular properties, with OSI, tuning width and DSI being not significantly different between Emx1:Cre-miR-132/212$^{-/-}$ mice and their age-matched littermates expressing exclusively the floxed *miR-132/212* allele (miR-132/212$^{fl/fl}$ mice) or the Cre-recombinase allele (Emx1:Cre-wt mice; One-way ANOVA, $P = 0.167$, $P = 0.300$ and $P = 0.893$ respectively; Fig. 6c–e). In contrast, binocular orientation matching appeared to be significantly impaired in Emx1:Cre-miR-132/212$^{-/-}$ animals (One-way ANOVA $P < 0.01$, *post hoc* Holm-Sidak test $P < 0.01$; Fig. 6f,g), thus recapitulating the phenotype of ubiquitary null mice. Analysis of average binocular matching of single mice confirmed the presence of an impairment exclusively in Emx1:Cre-miR-132/212$^{-/-}$ mice (One-way ANOVA $P < 0.05$, *post hoc* Holm-Sidak test, miR-132/212$^{fl/fl}$ versus Emx1:Cre-miR-132/212$^{-/-}$ $P < 0.05$; Emx1:Cre-wt

versus Emx1:Cre-miR-132/212$^{-/-}$ $P < 0.05$; Emx1:Cre-wt versus miR-132/212$^{fl/fl}$ $P = 0.808$; Supplementary Fig. 7). These data indicate that the action of miR-132 family in excitatory forebrain cells is required for the normal development of binocular matching of orientation preference in the primary visual cortex.

## Discussion

During the first month of postnatal life the mouse visual cortex undergoes dramatic morphological and functional changes that leads to adult-like neuronal receptive fields[17] and the maturation of visual function[18,41]. We show that these changes are paralleled by a considerable transcriptome rearrangement: many miRNAs undergo a remarkable change of expression between P10 and P28 with the top hits being miR-29, that was previously involved in regulating epigenetic enzymes important for cortical plasticity[6]; miR-338 and miR-219, that were suggested to be key players in myelination[42,43]; and miR-132/212, a miRNA family previously involved in synaptic plasticity[8–11,26,44–47]. The combined analysis of miRNAs and mRNAs in the visual cortex of wt and *miR-132/212* null mice revealed that genes downregulated with age and

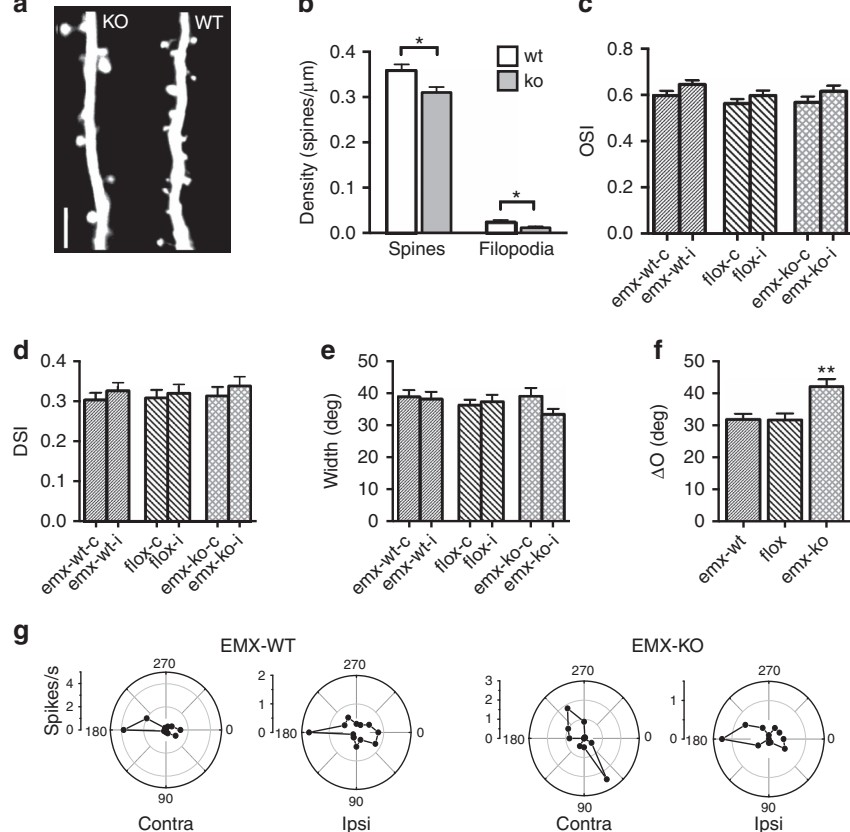

**Figure 6 | Deletion of *miR-132/212* in excitatory neurons reproduces the binocular matching deficit of null mice.** (**a**) Image of a dendritic branch from wt and miR-132/212 null (ko) mice at postnatal day P30, showing decreased spine density. Calibration bar: 1 μm. (**b**) Spine (sp) and filopodia (fi) density in ko ($N = 13$ mice, 50 dendrites) and wt mice ($N = 13$ mice, 48 dendrites). (**c**–**e**) No difference in orientation selectivity (**c**), orientation tuning width (**d**) and direction selectivity (**e**) is present in mutants (emx-ko, $N = 5$ mice, $n = 119$ cells) compared to control Emx:Cre (emx-wt, $N = 6$ mice, $n = 156$ cells) and floxed animals (flox, $N = 6$ mice, $n = 137$ cells). (**f**) At P30 emx-ko have a higher ΔO with respect to flox and emx-wt age-matched littermates. (**g**) Polar plots of orientation-tuned responses from contralateral and ipsilateral eye for emx-wt (up) and emx-ko (bottom) are depicted. Asterisks indicate statistical significance. Error bars represent s.e.m.

upregulated by miR-132/212 deletion are highly enriched with miR-132-3p targets and, to a much lesser extent, with targets of miR-212-5p, the only two members of the miR-132 family abundantly expressed in the visual cortex. Taken together, these data demonstrate that miR-132 family contributes to shape the developmental regulation of visual cortex transcriptome and prompted us to analyse the effects of genetic deletion of the *miR-132/212* locus on functional development of the visual cortex. We observed a specific deficit in the maturation of binocular matching of orientation preference in neurons of the binocular visual cortex of mutants accompanied by a remarkable impairment in depth perception.

OD plasticity is one the most studied models to understand how experience regulates brain development. A flurry of molecular mediators have been shown to be regulated by visual experience and involved in OD plasticity[17,27]. However, very little is known about the role of molecular factors involved in OD plasticity in normal visual development of non-deprived animals. A reasonable expectation would be that in absence of these factors visual development should be dramatically impaired. However, many studies have shown that functional development of cortical units is a multi-faceted process involving experience-independent and experience-dependent aspects occurring at specific time windows of development[14,17,48]. Thus, it is difficult to predict which features could be regulated by factors mediating experience-dependent plasticity. Recent work showed that a late

occurring developmental process is the binocular matching of orientation preference[19–23]. This process occurs in coincidence with the rise of miR-132 expression in the visual cortex[8], with simple cells reaching adult levels of binocular matching of orientation preference before complex cells[23]. Importantly, this process is also temporally coincident with the CP for OD plasticity[19] and is disrupted by MD[19] suggesting that mechanisms underlying OD plasticity in deprived mice might overlap with those involved in development of binocular matching of orientation preference. Our data support this possibility by showing that the absence of *miR-132/212*, that resulted in no OD plasticity, was associated with a specific impairment of binocular matching of orientation preference. By contrast, early developing features of cortical neuron RF, like orientation and direction selectivity, were unaffected. Moreover, the effects of MD on binocular matching were occluded by *miR-132/212* deletion suggesting that MD might act on binocular matching by reducing miR-132/212 levels. This hypothesis is based on previous data showing that miR-132 and its primary precursor are strongly downregulated by MD (refs 8,9). A possible scenario emerging from these observations is that visual experience after eye opening endows cortical cells with plasticity mechanisms, such as the post-transcriptional target regulation by miR-132/212, necessary for refinement of binocular connections onto visual cortical neurons. These mechanisms would overlap with those involved in changing OD in MD animals. MiR-132 family is expressed in

an activity-dependent manner also in the monocular visual cortex[8] therefore it seems unlikely that these miRNAs are specifically dedicated to the formation of binocular cells. A more likely possibility is that miR-132 family could be involved in experience-dependent processes occurring also in this region[14] at the age when miR-132/212 is expressed.

Our data show that *miR-132/212* mutants display an impaired maturation of binocular depth perception revealed using the visual cliff test that relies on binocular vision. This is the first time that an impairment of binocular matching of orientation preference of cortical neurons has been related to a behavioural failure, suggesting a physiological requirement of this neuronal feature in mouse vision.

We can exclude the contribution of non-visual cues to the preferential exploration of the shallow side because when we placed visual stimuli at the same height immediately below the glass plates, we did not detect any difference in the exploration of the two sides of the apparatus. In addition, the effect reported was not due to abnormal levels of anxiety or general exploratory activity in mutant mice as indicated by the equal time spent in the central and peripheral portion of the apparatus, and the comparable path length and locomotion velocity shown by wt and *miR-132/212* null mice. Finally, the similar VA of wt and mutant mice excludes the possibility that the different performance in the visual cliff test was due to blurring of the patterned visual stimuli in mutants.

It is well-known that depth perception exploits multiple visual cues, however binocular cues, such as disparity, are known to be particularly important also in mice[49]. It has been proposed that the response of visual cortical neurons to the inputs from the two eyes need to be tuned to similar orientations to encode binocular disparity of stimulus phase[19,50], suggesting that impaired matching of binocular orientation preference induced by *miR-132/212* deletion underlies the defective depth perception of mutants.

Our data also show that the impairment of binocular matching and depth perception due to *miR-132/212* deletion is comparable to that observed in MD wt mice. By contrast, OD of cortical neurons is differentially affected by these two manipulations, suggesting that depth perception preferentially reflects the state of binocular interactions rather than the strength of the inputs of the two eyes onto cortical neurons.

How could the transcriptional regulation of *miR-132/212* cluster control experience-dependent development of functional properties of visual cortical neurons? A series of *in vitro* and *in vivo* studies revealed that miR-132 family has an active role in brain structural plasticity. Indeed, miR-132 has been flagged as an important regulator of activity-dependent shaping of dendritic morphology and arborization, and spine density[8,46,51] acting through the activation of the Pac1-PAK-actin remodelling pathway[47,52]. Moreover, miR-132 was found to mediate synaptic plasticity in the hippocampus[53,54] and in the cortex[26], and alteration of its expression has been documented in several neuropsychiatric disorders[55,56]. Most importantly, among the miR-132 target genes downregulated with age in wt mice, but remaining at significant high levels in *miR-132/212* null mice, there are genes potentially involved in binocular matching like MeCP2 (ref. 21), the mediator of brain-derived neurotrophic factor (BDNF) signalling Sos1 (ref. 23), and phosphatase and tensin homolog (PTEN), an antagonist of the plasticity and growth mTOR/Akt pathway[30].

In conclusion, our study represents the first characterization of a miRNA role from its *in vivo* transcriptional and target regulation to its moulding action on cortical receptive fields and on its behavioural consequences on vision. Many microRNAs were found to be strongly regulated with age suggesting that they

might contribute to the molecular regulation of the cortical transcriptome and eventually to functional development. Their number is likely to be underestimated by our study considering that cell-specific analyses could reveal additional regulated miRNA with highly selective expression. Thus, we surmise that miRNAs likely represent a novel molecular layer of control of postnatal cortical development. The pathogenetic relevance of miR-132 regulated processes during development is further supported by the presence of brain disease related genes among the developmentally regulated miR-132 targets altered in the *miR-132/212* mutant including *MeCP2* (ref. 57), *Pten* (ref. 58), Ras-regulating genes *Sos1* and *Rasa1* (ref. 59), *Mmp16* (ref. 60), *Runx1t1* (ref. 61), *Sox11* (ref. 62), *Sox5* (ref. 63) and *Gpd2* (ref. 64). Since many neurodevelopmental disorders including autism are related to alterations of neuronal connectivity and synaptic plasticity, miR-132 dysregulation and subsequent abnormal expression of miR-132 target genes could contribute to some pathological traits present in these diseases. Thus, a strategic modulation of miR-132/212 expression may offer a new therapeutic approach for these severe disorders.

## Methods

**Animals.** Mice were handled according to protocol approved by the Italian Ministry of Health (Authorization number 147/2014-B) and commensurate with NIH guidelines for the ethical treatment of animals. All mice (C57BL6J background of both sexes) were kept in a normal 12 h light/dark cycle in a temperature-controlled room (20 °C) with *ad libitum* access to food and water *ad libitum*. Mice with ubiquity deletion of *miR132/212* and wt littermates were used[26]. To target *miR132/212* deletion to excitatory neurons in the forebrain we used a mouse[40] expressing Cre recombinase under the Emx1 promoter (Emx1:Cre). MiR132/212[+/fl] male mice[26] were crossed with Emx1:Cre females to generate a mouse line carrying the floxed *miR132/212* and Emx1:Cre alleles. We used this specific breeding strategy to avoid that undesired germ-line recombination could occur in the Emx1-Cre testis, a tissue known to express Emx1 (ref. 65). Mice of three genotypes were used as experimental animals: mice carrying the forebrain specific deletion of *miR-132/212* (Emx1:Cre-miR-132/212[−/−]), mice expressing the floxed allele but not Cre-recombinase (miR132/212[fl/fl]), and mice expressing Cre-recombinase but not carrying the floxed allele (Emx1:Cre-miR-132/212[+/+]). These genotypes were obtained in the same litter by crossing male miR132/212[+/fl] male mice with female Emx1:Cre-miR-132/212[+/fl] females. Routine tail genotyping of these mice was carried out using 3 primers resulting in bands of 373 bp for the wt allele, 420 for the floxed allele and 550 for the deleted allele[26]; only mice in which germ-line recombination could be excluded because of the presence of the 420 bp floxed band in the tail tissue were analysed.

**RNA sequencing.** RNA libraries were constructed from the samples extracted from the 12 replicates in the three groups P10-wt, P28-wt and P28-KO using the TruSeq RNA Sample preparation kit v2 (Illumina) and sequenced (2 × 100 bp reads, about 30 M reads/sample) on a Illumina Hiseq2000 platform at the Istituto di Genomica Applicata (Udine, Italy). The resulting sequencing data for each library were post-processed to produce FastQ files, then demultiplexed using Illumina software CASAVA 1.8.2. The number of sequencing reads obtained for each replicate is reported in Supplementary Fig. 1. The quality of the sequences was further assessed using the PHRED quality scores produced in real-time during the base-calling stage of the sequencing run. The mean PHRED score by sequencing cycle was computed separately for each group of replicates.

The reads from each replicate experiment were separately aligned to the reference genome mm10 and corresponding transcriptome using the short-read aligner ELAND v2e (Illumina). The RefSeq gene annotations were used to define the set of known exons and splice junctions. Reads uniquely aligned to known exons or splice junctions with no more than two mismatches within at least one 25 bp segment of the sequencing reads were included in the transcriptome. Reads uniquely aligned but with more mismatches, or reads mapping several locations in the reference genome, were filtered out to use only unambiguously located reads in the next steps of the RNA-seq analysis. Gene expression levels were directly computed from the read alignment results for the 12 replicates in the groups P10-WT, P28-WT and P28-KO. Standard RPKM values (reads per kilobase of exon model per million mapped reads) were extracted for each gene covered by the sequencing data and each replicate used in this study. Differential transcriptional analyses were performed using CyberT (ref. 66), upgraded to handle RNA-seq data[67], across each pair of groups. A sliding window size of 101 genes and a confidence value of 10 were used as bayesian-regularization parameters across all pair-wise comparisons. To correct for multiple comparisons and the multiple hypothesis testing problem, we added the posterior probability of differential expression that is computed from the distribution of the raw *P* values, by fitting a

mixture of beta distributions to this distribution, and then computing the probability that the observed $P$ value belongs to the component in the mixture associated with differentially expressed genes[68]. Genes of interest having a posterior probability of differential expression above 0.6 are considered significant. Functional enrichment was performed for each of the differentially expressed gene lists generated in the differential analysis. For KEGG pathway analysis of our gene set data, we used the software GeneCodis (http://genecodis.cnb.csic.es/analysis).

MiRNA target prediction was performed using existing database scoring the strength of the interaction (Target Scan Mouse 7.1). Genes with only poorly conserved sites were not included.

Similar data were obtained using an alternative method to identify targets combining three different sources:

(1) TargetScan: the 'Broadly conserved' category of data sets corresponding to conservation among vertebrates was used. The conservation score used by TargetScan is PCT.

(2) microRNA.org: targets were identified from target sites found by MiRanda with a threshold miRSVR score ($\leq -1.2$). It is possible to identify non-conserved sites in this database. However, since we take the intersection of sources, the site was found to be conserved in either TargetScan or using BBLS.

(3) miRanda and BBLS: targets are identified from target sites which were found by MiRanda and filtered on a BBLS $\leq 1.0$ from mm10.

To show if the miRNA target lists were found to be enriched in gene lists deriving from P10-wt versus P28-wt, and P28-wt versus P28-KO comparisons we analysed downregulated gene targets for those microRNA upregulated in the same pair-wise comparisons as well as vice versa. Again, a Fisher exact test was used to compute the statistical likelihood of enrichment for each microRNA's target list.

**In vivo electrophysiology.** For surgery, mice were sedated with isoflurane (Forane, 3%) followed by urethane (0.7 g kg$^{-1}$, i.p., at 20% w/v in ringer). Dexamethasone (Soldesam, 2 mg kg$^{-1}$) was administered subcutaneously to reduce secretions and oedema. The animal was placed on a stereotaxic apparatus and maintained at 37.5 °C by a feedback-controlled heating pad. The animal was also ventilated through an oxygen mask. During surgery, eyes were protected by appying a dexamethasone-based ointment (Tobradex, tobramycin 0.3% and dexamethasone 0.1%), then with a thin layer of silicon oil. Local anaesthesia was obtained with a subcutaneous injection of lidocaine (Angelini). After exposing the skull, we performed a small craniotomy (2 mm in diameter) over the binocular visual cortex (2.8–3.3 mm lateral and 0.1 mm anterior to lambda) living dura mater intact. An electrode (2 × 2-tet-3 mm-150-150-121-A16-15, Neuronexus Technologies) was slowly lowered into the cortex to an appropriate depth to record local field potentials and single-unit activity and was allowed to settle for 30–40 min before the beginning of recordings. The recording sites were located between layers III and V, therefore the physiological data recorded mostly reflects the properties of these layers. At the end of the experiment the animal was killed by overdose with of urethane. Signals were acquired using 16 channels Neuralynx device and data analysis was performed using a custom software written in Matlab. Visual stimuli were generated in Matlab using the Psychophysics Toolbox extension and displayed with gamma correction on a monitor (Sony Trinitron G500, 60 Hz refresh rate, 32 cd m$^{-2}$ mean luminance) placed 20 cm from the mouse, subtending 60–75° of visual space.

Local field potential- We measured the contralateral to ipsilateral ratio of VEPs to measure OD plasticity. Extracellular signal was filtered from 0.3 to 275 Hz and sampled at 30.3 kHz. VEPs in response to square wave patterns with a spatial frequency of 0.03 c/deg and abrupt phase inversion (1 Hz temporal period), were evaluated in the time domain by measuring the P1 peak-to-baseline amplitude and latency. We used computer controlled mechanical shutters to collect data from each eye, reducing possible effects due to changes in behavioural states, and adaptation. VA was assessed using different spatial frequencies (in 6 logarithmic steps between 0.03 to 0.96 c/deg) in pseudorandom order. VA was obtained by extrapolation to zero amplitude of the linear regression through the data points in a curve where VEP amplitude is plotted against log spatial frequency. Baseline response (0.96 c/deg) was subtracted from raw amplitude.

single-unit activity—for single-unit recording extracellular signal was filtered from 0.6 to 6 kHz. Sampling rate: 30.3 kHz. Spiking events were detected online by voltage threshold crossing and waveforms of 1 ms were acquired around the time of threshold crossing. To improve isolation of units, recordings from groups of four neighbouring sites (tetrode) were linked, so that each spike was composed by 4 waveforms. Data were loaded on OffLine Sorter software (Plexon) and principal component analysis was performed to score spikes with an high degree of similarity in a 3D feature space, where dimensions correspond to the first 3 principal components. Waveforms from each electrode of the tetrodes were processed together to improve isolation. Clusters were progressively defined using convex hulls and then recalculating principal component analysis. Quality of separation was determined based on the following criteria: (1) during manual clusterization with convex hulls, raw waveforms in the clusters were visually inspected to check the uniformity of single waveforms; (2) clusters contained <0.1% of spikes within a 1.0 ms interspike interval; (3) auto- and cross-correlograms of the clusters were also inspected to reveal if the cluster contained more than a single unit or if several clusters contained spikes of the same unit; (4) the peak amplitude of a unit remained stable over the entire recording session. Units were included in the

sample for analysis of tuning properties when they had an average peak firing rate, across trials of the optimal stimulus for the dominant eye, of >0.5 Hz.

Drifting sinusoidal gratings were used as visual stimuli (1.5 s duration, temporal frequency of 2 Hz, 12 different orientations with a step of 30°, 6 spatial frequency: 0.01, 0.02, 0.04, 0.08, 0.16 and 0.32 cpd). Stimulation was repeated five times per eye, with stimulus conditions randomly interleaved, and two grey blank conditions (mean luminance) were included in all stimulus sets to estimate the spontaneous firing rate. The average spontaneous rate for each unit was calculated by averaging the rate over all blank condition presentations. Responses at each orientation and spatial frequency were calculated by averaging the spike rate during the 1.5 s presentation and subtracting the spontaneous rate. The preferred stimulus was determined finding the combination of spatial frequency and orientation that maximize the response, independently for each eye. Orientation tuning curves were constructed for the spatial frequency that gave maximal response at this orientation. Given this fixed preferred orientation (OPref), the tuning curve was fitted as the sum of two Gaussians centred on OPref and OPref$+\pi$, of different amplitudes but equal width, with a constant baseline B. From this fit, we calculated two metrics: an OSI representing the ratio of the tuned versus untuned component of the response, and the width of the tuned component. OSI was calculated as follows: (respOPref-respOOrtho)/(respOPref + respOOrtho), where resp is the maximal response evoked by visual stimulation and OOrtho is the orientation orthogonal to the preferred one. Tuning width is the half-width at half-maximum of the principal gaussian. In addition, DSI was calculated as: (respOPref-respOOppo)/(respOPref + respOppo). The difference in preferred orientation between the two eyes (binocular matching, $\Delta O$) was calculated by subtracting ipsilateralOPref from contralateral OPref along the 180° cycle. ODI was calculated as follows: ODI = (respContra-respIpsi)/(respContra + respIpsi), Contra is the contralateral eye and ipsi is the ipsilateral eye. Simple and complex units were classified using *mtspectrumpb* from chronux toolbox[69]. Linearity of response was calculated at the orientation and spatial frequency that gave maximal response. We binned the 1.5 s presentation into a poststimulus time histogram at 100 ms intervals and subtracted the spontaneous rate. We then applied the discrete Fourier transform and computed F1/F0, the ratio of the first harmonic (response at the drift frequency) to the 0th harmonic (mean response). Units were then classified as narrow or broad spiking on the basis of properties of their average waveforms. The parameters used for discrimination were: the height of the positive peak relative to the initial negative trough, the time from the minimum of the initial trough to maximum of the following peak, and the slope of the waveform 0.5 ms after the initial trough. Two linearly separable clusters were found, corresponding to narrow-spiking (putative inhibitory) and broad-spiking (putative excitatory) neurons. These clusters were separated identically by both k-means and linkage clustering.

**Visual cliff test.** A modification of the published visual cliff[38] was employed. The apparatus consisted of a rectangular arena (50 × 36 cm) constructed in poly(vinylchloride) with white walls and bordered by black curtains to prevent the animal's escape. The arena was divided into two 25 × 36 cm Plexiglas plates. A moving platform, the depth of which could be varied by means of a mechanical scissor jack, was placed below each glass plate. A patterned floor consisting in 3-cm black and white checked photographic paper covered the platform's surface (Fig. 5a). Incandescent lamps placed below the two patterned floors illuminate both surfaces to equate the brightness of the two sides. A telecamera was hanging on the apparatus and was connected to a computer by which the experimenter could observe and record the animal's behaviour. Testing took place in a quiet room. The arena was divided into a shallow and a deep side. At the shallow side the patterned floor was positioned immediately below the glass plate, while at the deep side the checked platform was moved to 29 cm below the glass plate. Each animal was placed on the shallow side and the total time the mouse spent exploring each side of the arena was recorded. The trial ended after 5 min. The position of the shallow and the deep side of the arena was randomized between animals. The exploration path and velocity of each animal were recorded and analysed with the Noldus Ethovision system. The arena was cleaned between trials with a 10% alcohol solution. To rule out that non-visual cues could affect the output of the task, we also assessed animal behaviour in the absence of visual cues, with visual stimuli placed immediately below the glass plates of the apparatus.

**Spine density assessment.** To study spine density, miR-132/212 transgenic and wt mice crossed with the mouse line expressing green fluorescent protein in layer V pyramidal neurons (line M (ref. 70)) were perfused with 4% paraformaldehyde and brains were post-fixed for 2 hours and incubated in 30% sucrose in PBS O/N. Coronal sections (200 μm) were cut on a vibratome. Slices were mounted on glass slides and stacks of images were acquired by using a Zeiss ApoTome microscope. A low magnification image was first acquired to guide the experimenter to acquire a high magnification image in V1 based on anatomical points of reference. Spines were manually counted using ImageJ.

**Data availability.** The sequences generated and analysed during the current study are available in the GEO repository (https://www.ncbi.nlm.nih.gov/geo/query/acc.cgi?acc=GSE95649).

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

## Acknowledgements

This work has been supported by Epigenomics Flagship Project EPIGEN MIUR-CNR and Telethon project GGP1114. We thank Dr Guido Marco Cicchini and Prof Alessandro Cellerino for useful discussion of the results.

## Author contributions

N.C., C.M., D.N., P.T., J.T. and P.B. produced and analysed molecular data. R.M. and G.C. produced and analysed electrophysiological data, L.B. and D.S. produced and analysed behavioural data. G.D.S. and E.P. provided and analysed histological data. R.M., L.B. and T.P. wrote the manuscript with help from co-authors. Each author read and approved the final manuscript. T.P. supervised all aspects of the project.

## Additional information

**Competing interests:** The authors declare no competing financial interests.

