## [Peer Review File · Nature Communications]

Reviewers' comments:

Reviewer #1 (Remarks to the Author):

The paper by Mazziotti and colleagues addresses the role of a specific class of miRNAs (miR-132/212) on visual cortex development and plasticity. As had been shown earlier, these miRNAs are upregulated during visual cortex development, and they play an important role for the regulation of OD plasticity during the critical period. The authors now extend these findings by demonstrating a large overlap in the targets of miR-132/212 with transcripts that are downregulated over this developmental period. In line with this, many of these same genes are upregulated in miR-132/212 ko mice. They go on to show that miR-132/212 ko mice display a specific developmental deficit: while neurons in the visual cortex show normal orientation and direction selectivity, the binocular matching of the preferred orientation, which normally happens during the critical period, does not take place. Importantly, the authors also show that this deficit has behavioral consequences, in that these mice show impaired depth perception in the visual cliff test. Finally, they demonstrate that this defect does also occur when these miRNAs are specifically deleted from forebrain excitatory neurons only.

This is a very nice study that goes beyond earlier work on these miRNAs by demonstrating that they play a crucial role for orchestrating a specific aspect of the normal, experience dependent development of cortical connectivity, which then is also important for a specific sensory capability, namely depth perception.

I have only a few minor comments:

- I would like to learn in a bit more detail based on which criteria recorded cells were included or not included in the sample (signal to noise, spike rate, OSI cutoff...).
- Likewise, the authors are very brief on their spike sorting: How sure can we be that all the data presented are from single units?
 - Out of interest: was there any systematic interocular difference in preferred orientation in the ko mice (and actually also the wt mice)? I.e. not plotting the absolute value of the difference but the actual difference.
- In their introduction the authors list direction selectivity as one feature that develops without visual experience. While this might be true for the mouse, data from the Fitzpatrick lab very clearly show that this is not the case in higher mammals.
- Numbers of mice should be given together with the number of cells
- The acronym "KEGG" should be explained.
- Results, p10, line 232: please give percentage for wt mice
- Discussion, p19, line 459: Parker, 2007 does not report any mouse data
- Fig. 1a: please explain red and black dots

Reviewer #2 (Remarks to the Author):

The present manuscript by Mazziotti et al., 'Mir-132/212 shapes age-dependent transcriptome and is required for binocular matching of orientation preference and depth perception' examines the role of the microRNA 132 in the experience-dependent development of visual circuitry. The major findings reported in the study are that genetic deletion of miR-132 impairs ocular dominance plasticity, depth perception as measured with an original behavioural assay, and spine density.

MiR-132 has been implicated over the last decade in several aspects of cortical development. This work is cited extensively in the Discussion. In fact, two papers were published in Nature Neuroscience in 2011 ascribing a role to miR-132 in ocular dominance plasticity (ODP). The papers, although focused on the same microRNA reached differing conclusion. Mellios et al, from the Sur lab, reported that sequestering miR-132 impaired ODP. In contrast, the Tognini et al., from the Pizzorusso lab,

sharing many of the same authors as this manuscript, reported that preventing the modest reduction in miR-132 during monocular deprivation (MD) also prevented ODP. Thus, unexpectedly, either reducing or augmenting miR-132 function appears to disrupt experience-dependent visual plasticity. Here the authors examine whether genetic deletion of miR-132 disrupts ODP (as published by Mellios et al.) and another perhaps related but definitely interesting form of visual plasticity, binocular matching of orientation preference.

The originality and interest of this study are somewhat limited. MiR-132 has already been implicated in visual plasticity. The present manuscript makes no attempt to reconcile how either up-regulation or down-regulation of this transcript can yield the same result. Instead, the manuscript redirects to a related form of visual plasticity and reports outcomes that are largely incremental extensions of published work. Overall, the magnitudes of the effects reported for differences in binocular matching, depth perception, and spine density are modest if not questionable. The significance of the present work is moderate and further dampened by pervasive technical concerns.

There are substantive problems with the experimental design and interpretation for each figure. Below, specific major concerns are presented for each figure. The statistical analysis appears to potentially be correct in most circumstances. However, in several instances, the authors have opted to examine differences in the mean from units gathered from multiple mice. This approach is inferior to comparing the averages between mice.

Critically, the differences reported between genotypes are of smaller magnitude than the pervasive and significant discrepancies with these same techniques presented (and reproduced) relative to the relevant literature on both binocular matching and stereopsis. Overall, this manuscript fails to reproduce essential changes in binocular matching reported by several groups in multiple papers that established this line of inquiry. Thus, most of the conclusions presented in the manuscript are not supported by good evidence.

There are numerous mis-statements in the manuscript and incorrect referencing. For example: 'when mouse visual function reaches adult levels of performance (P28)' (Cancedda et al., 2004; Hoy and Niell, 2015; Kang et al., 2013; Ko et al., 2013; Rochefort et al., 2011; Wang et al., 2010) - This assertion that P28 mice display adult levels of performance is not supported by any of the listed references and is refuted by the results presented in each study.

Cancedda et al, 2004 report acuity at P28 is only 80% of that at P60
Hoy and Niell, 2008 examine mice at 1-7 days after eye-opening and P60+, not P28
Kang et al, 2013 examine mice at P30-36
Ko et al, 2013, examine mice at P22-26. They never state the parameters they examine reflect adult performance, rather than they are greater than at P12-14, a younger age for comparison.
Rochefort et al, 2011 report continued maturation of visual circuitry until at least 2 months of age
Wang et al, 2010 report substantive changes in binocular matching between P23 and P31.

Below are major concerns for each Figure.

Figure 1.

A. Given that the authors previously published a study in Nature Neuroscience reporting that it was the decrease in miR-132 expression that is required for ODP, it is confusing that here the authors chose to compare transcript expression at P10 vs. P28, and P28 KO vs. WT. What is the reader meant to deduce from this list of pathways spanning axon guidance, synapse formation, and activity-dependent formation? These pathways are established contributors to experience-dependent cortical plasticity. This figure is simply not informative.

B. The authors should list the 39 genes at the center of the Venn diagram, the magnitude of the effect and confirm these findings by immunoblot for at least the top 10.

Figures 2

A. The OSI (.6 here vs. \sim .9), DSI (.35 here vs. \sim .2) and OS tuning width (35 here vs. \sim 22) differ substantially and significantly from the adult values reported elsewhere, including a detailed study by the Niell lab, Hoy et al. 2015. Given that the authors state in the methods that they are using the same approach as the Neill and Stryker, 2008, this discrepancy must be resolved. The likely reason is that the authors are not examining mice with mature physiologic properties despite their assertion (l. 86). This choice of experimental design attenuates interpretation of the experiments presented in the manuscript.

Figure 3

A. The deltaO presented here for WT mice conflicts with published reports. These discrepancies are not mentioned or explored. Here the values for WT mice are reported as \sim 30 degrees. Wang et al, Neuron 2010 report values closer to 17 degrees. This Neuron paper established this line of inquiry. The values reported here are beyond the range for the normal developmental trajectory but similar to those for dark-reared mice. Given these problems, it is unclear that the magnitude of difference reported by the authors is stochastic variation because the authors fail to report the number of mice examined. For comparison, Wang et al. examined 15-18 mice per group.

B. The impaired binocular matching is a fairly interesting result but there are extensive concerns that this finding may not be specific for miR-132, but a general consequence of transcriptional mis-regulation. This possibility is not raised or addressed in the manuscript. This should be addressed experimentally.

Minor:

i. the number of units and mice examined should be included in the figure legend as well as the statistical test employed.

Figure 4

A. Panels A – C. The finding that neutralizing miR-132 impairs ODP was published in 2011 with calcium imaging in vivo (Mellios et al., 2011). That the authors can reproduce this finding by genetic deletion of miR-132 is an incremental addition.

B. Panel B. The authors should present the ODP as contralateral bias index per mouse, not ODI values combined for all units across all mice. CBI is the standard in the field, and has been employed by the Pizzorusso lab several publications.

C. Panel D. Again, the values for deltaO reported here differ significantly from other studies. Moreover, the magnitude of the effect is far less here. Wang et al report that MD prevent the decrease in deltaO from 30 degrees to \sim 17 degrees, nearly a halving of deltaO. That is twice the magnitude of effect reported here. Whether these experiments have been performed and/or analyzed properly is a pervasive concern.

Minor:

i. 'Single-unit analysis of the ocular dominance index (ODI; 259 (Niell and Stryker, 2008) confirmed the lack of OD plasticity in null mice'

- This reference is incorrect. Niell and Stryker do not examine ODP or introduce ODIs. ODIs are typically presented for experiments employing intrinsic imaging, such as Cang et al. 2005.

Figure 5.

A. This assay seems unlikely to report depth perception. This original work on depth perception in mice (Fox, 1965) revealed that mice prefer the shallow side to the deep side 90% of the time (10 mice). That corresponds to a DI, the metric presented here, of 0.8, TWICE that reported by the authors. Converting the DI of 0.4 to percent preference corresponds to a preference of the shallow side in this study of 70%. Chance distribution of the two sides is 50%, yet the p value is < .001 for ten mice by the analysis presented by the authors. This seems dubious.

- The data should be presented and the average percent time each mouse dwells on the shallow side. A data point should be presented for each mouse. The stats should be recalculated.

B. The principal findings of this paper are that WT mice exhibit a preference of the shallow side of 70% while KO mice has a lower preference of only 60% (DI = 0.2). To be frank, this 10% difference is not compelling or interesting.

C. The DI for WT mice is 0.4 in panel E and almost 0.6 in panel H, essentially the same magnitude of difference the authors assert separates WT and KO mice. Some explanation is required.

D. The number of approaches to the center point differs dramatically between genotypes. This assay is likely reflecting a combination of anxiety, exploration and other cues. In addition, given that the mice can palpate the glass surface with their whiskers, a definitive perception, it is questionable that this assay reports depth perception.

Figure 6.

A. Are these neurons in V1? What is the evidence these neurons are in V1? It is widely appreciated that EGFP-M does not express GFP in V1 (Holtmaat, 2005). Hofer et al, Nature 2009 screened scores of mice to find a handful that expressed a few neurons in V1. How have the authors made this determination? Some verification must be presented

B. Panel A. This image quality is substandard. In addition, it does not represent the spine density reported in panel B. How this image has been processed must be detailed in the methods section, particularly any non-linear modifications.

C. Panel B reports more than 1.5 spines per micron for L5 neurons. This value is 3X that reported elsewhere including Hofer, 2009 (see Fig. 3a). This spine density is dramatically higher than the bulk of the available literature and raises substantive concerns about the accuracy of this analysis.

D. Panel B, the magnitude in the difference in spine density is approximately 10%. This is not convincing, particularly given the small sample examined and the quality of the representative images.

E. Panels C-D suffer from many of the same shortcomings as Figures 2 and 3. In addition, the authors must provide a demonstration that they can detect recombination of the miR-132 locus because EMX1:Cre displays germline recombination at high frequency.

Reviewer #3 (Remarks to the Author):

It was a pleasure to read the manuscript by Mazziotti et al. The manuscripts detail the investigation of small RNA expression and posttranscriptional regulation in the visual cortex of the developing mouse.

The authors identified miR-132/212 as being both altered and influential on the transcriptome among other miRNA, many of which have been reported previously in other brain regions. This miRNA family is also known for its activity-associated expression pattern and was clearly an important target for further functional characterisation. To achieve this objective the authors used a combination of global and tissue specific knockout models to explore the functional significance in visual processing. This part of the manuscript is outstanding.

The authors were quick to focus on miR-132/212, but I was left wondering about the influence of other developmentally regulated miRNA? These miRNAs were way down the list of developmentally altered miRNA both by fold change and p-value, with miR-29 and miR-219 for example, showing much greater change. I could not see the qPCR validation of miR-132 or 212 but this data was available for miRs 29 and 219? Incidentally, the corrected p-values (or FDR) should also be provided for each of the tabulated miRNA and transcripts. Perhaps more detail on the rationale for this selection could also be provided. One wonders if this was just a convenient introduction for established animal models and experiments? It would be good to have the changes and transcriptional influences of the other developmentally regulated miRNA tabulated with their transcriptional effect size/odds ratios presented so this speculation can be dismissed. I also wonder how many target genes altered during development of the visual cortex were influenced by multiple miRNA. Would it be possible for the authors to map the network architecture of posttranscriptional interaction during development of the visual cortex? How were miRNA targets predicted? It would be nice if the quality or strength of the putative interaction were tabulated. Are these conserved targets or have non-conserved targets been included?

I would like to have seen more cross-referencing to previous studies of mammalian cortical development. These miRNA all seem familiar to me but it would be nice to see their correlates in other cortical structures. Are there any specific to the visual cortex or is there a difference in the levels or timing? Have any of these been associated with neuropathology?

It would be nice to have a bit more explanation of terms used in the figure legends. As an outsider to the field I had to refer back to the text to decipher the results in each panel.

The authors speculate on the target genes driving the miR-132 /212 associated changes in visual function. Would it be possible to recapitulate or rescue the miRNA-associated phenotype in vivo, by directly modulating some of these target genes?

REVIEWER #1

This is a very nice study that goes beyond earlier work on these miRNAs by demonstrating that they play a crucial role for orchestrating a specific aspect of the normal, experience dependent development of cortical connectivity, which then is also important for a specific sensory capability, namely depth perception.

We very much appreciate the reviewer for his/her interest in our manuscript.

I have only a few minor comments:

- I would like to learn in a bit more detail based on which criteria recorded cells were included or not included in the sample (signal to noise, spike rate, OSI cutoff...).

Units were included in the sample for analysis of tuning properties when they had an average peak firing rate, across trials of the optimal stimulus for the dominant eye, of >0.5 Hz. No other cutoffs were used. This information has been included in the Supplementary methods.

- Likewise, the authors are very brief on their spike sorting: How sure can we be that all the data presented are from single units?

Data were loaded on "OffLine Sorter" (Plexon) and principal component analysis (PCA) was performed to score spikes with an high degree of similarity in a 3D feature space, where dimensions correspond to the first 3 principal components. Waveforms from each electrode of the tetrodes were processed together to improve isolation. Clusters were progressively defined using convex hulls and then recalculating PCA. Quality of separation was determined based on the following criteria: 1) during manual clusterization with convex hulls, raw waveforms in the clusters were visually inspected to check the uniformity of single waveforms; 2) clusters contained <0.1% of spikes within a 1.0 ms interspike interval; 3) auto- and cross-correlograms of the clusters were also inspected to reveal if the cluster contained more than a single unit or if several clusters contained spikes of the same unit; 4) the peak amplitude of a unit remained stable over the entire recording session. This part has been included in the Supplementary methods.

- Out of interest: was there any systematic interocular difference in preferred orientation in the ko mice (and actually also the wt mice)? I.e. not plotting the absolute value of the difference but the actual difference.

We have attached below the distribution of the interocular difference in wild-type and *miR-132/212* null mice mice, either non deprived or monocularly deprived. Despite the loss of binocular matching in mutants and in WT MD mice, no systematic differences in interocular preferred orientation were detected in any experimental group.

- In their introduction the authors list direction selectivity as one feature that develops without visual experience. While this might be true for the mouse, data from the Fitzpatrick lab very clearly show that this is not the case in higher mammals.

We restricted our statement to rodents deleting irrelevant literature.

- Numbers of mice should be given together with the number of cells

We added the number of mice and cells in legends.

The acronym "KEGG" should be explained.

Done.

- Results, p10, line 232: please give percentage for wt mice

Done.

- Discussion, p19, line 459: Parker, 2007 does not report any mouse data

We removed this quotation.

- Fig. 1a: please explain red and black dots

Done.

REVIEWER #2

MiR-132 has been implicated over the last decade in several aspects of cortical development. This work is cited extensively in the Discussion. In fact, two papers were published in Nature Neuroscience in 2011 ascribing a role to miR-132 in ocular dominance plasticity (ODP). The papers, although focused on the same microRNA reached differing conclusion. Mellios et al, from the Sur lab, reported that sequestering miR-132 impaired ODP. In contrast, the Tognini et al., from the Pizzorusso lab, sharing many of the same authors as this manuscript, reported that preventing the modest reduction in miR-132 during monocular deprivation (MD) also prevented ODP. Thus, unexpectedly, either reducing or augmenting miR-132 function appears to disrupt experience-dependent visual plasticity. Here the authors examine whether genetic deletion of miR-132 disrupts ODP (as published by Mellios et al.) and another perhaps related but definitely interesting form of visual plasticity, binocular matching of orientation preference.

The originality and interest of this study are somewhat limited. MiR-132 has already been implicated in visual plasticity. The present manuscript makes no attempt to reconcile how either up-regulation or down-regulation of this transcript can yield the same result. Instead, the manuscript redirects to a related form of visual plasticity and reports outcomes that are largely incremental extensions of published work.

A: To our knowledge, this is the first time in literature that an impairment of binocular matching of orientation preference of cortical neurons has been related to a behavioral impairment, suggesting a physiological requirement of this neuronal feature in visual perception. Moreover, investigating the role of factors important for ocular dominance plasticity in normal development has been performed very rarely, and, with the exception of Wang et al, 2013, only analyzing visual acuity. Furthermore, we provide the first genome-wide combined analysis of the transcriptome and the miRNome during development of the mouse cerebral cortex providing reference data for molecular studies in the field. We think that the novelty and the relevance of our data rely on these points.

The issue of the dosage effect of miR-132 in plasticity is certainly interesting, but it has been extensively studied: Hansen et al. employed a transgenic mouse model to show that an ~ 5-fold overexpression of miR-132 in excitatory forebrain neurons leads to significant deficits in recognition and spatial memory (Hansen et al., 2010). Consistent with this, lentiviral-based overexpression of miR-132 in the perirhinal cortex was found to reduce recognition memory capacity (Hernandez-Rapp et al., 2015). As with transgenic miR-132 overexpression animals, germline miR-132/212 knockout mice exhibited deficits in recognition and spatial memory (Scott et al., 2012). However, when Hansen et al. used a Tet-off system to titer transgenic miR-132 to levels that paralleled the levels observed following a learning paradigm (i.e. ~ 2-fold above basal), they found that cognitive capacity was enhanced (Hansen et al., 2013). These data are consistent with the idea that maintaining miR-132 within a limited/physiological range is also essential for visual cortex plasticity. Analysis of the transcriptome of miR-132 overexpressing and knockout mice in the hippocampus (Hansen et al., 2016) and in the visual cortex is ongoing to reveal how these dose dependent effects can be achieved.

Hernandez-Rapp J, Smith PY, Filali M, Goupil C, Planel E, Magill ST, et al. Memory formation and retention are affected in adult miR-132/212 knockout mice. *Behav Brain Res* 2015; 287:15-26.

Scott HL, Tamagnini F, Narduzzo KE, Howarth JL, Lee YB, Wong LF, et al. MicroRNA-132 regulates recognition memory and synaptic plasticity in the perirhinal cortex. *Eur J Neurosci* 2012; 36:2941-2948.

Hansen KF, Sakamoto K, Wayman GA, Impey S, Obrietan K. Transgenic miR132 alters neuronal spine density and impairs novel object recognition memory. *PLoS One* 2010; 5:e15497.

Hansen KF, Karelina K, Sakamoto K, Wayman GA, Impey S, Obrietan K. miRNA-132: a dynamic regulator of cognitive capacity. *Brain Struct Funct* 2013; 218:817-831.

Hansen KF, Sakamoto K, Aten S, Snider KH, Loeser J, Hesse AM, Page CE, Pelz C, Arthur JS, Impey S, Obrietan K. Targeted deletion of miR-132/-212 impairs memory and alters the hippocampal transcriptome. *Learn Mem.* 2016 23(2):61-71.

There are substantive problems with the experimental design and interpretation for each figure. Below, specific major concerns are presented for each figure. The statistical analysis appears to potentially be correct in most circumstances.

However, in several instances, the authors have opted to examine differences in the mean from units gathered from multiple mice. This approach is inferior to comparing the averages between mice.

A: We thank the reviewer for his/her thorough review addressing presentation of results and the need of more control experiments. We list below the most significant modifications made to the manuscript to address the reviewer's comments. Although the results or interpretation of results have not changed, we sincerely believe that the work and manuscript has been significantly improved:

- All the ephys data have been reanalyzed making comparisons across the averages between mice in addition to previous analysis of individual units. All the comparisons that were significant with the unit-based analysis were also significant with the subject-based analysis.
- We have added new data obtained from P60 mice showing that misalignment between orientation preference in adult wild-type mice is around 20 degrees as previously published, a value that is dramatically lower than the value measured in age-matched miR-132 mutant mice.
- We performed additional visual cliff experiments in which the visual cue was eliminated by placing the visual stimuli in the two sides of the apparatus at the same depth (just below the plexiglas floor). No preference for any of the two sides was present, eliminating the possibility that non-visual cues were involved in the preference for the shallow side displayed by wild-type mice in the visual cliff test. No difference between wt and ko emerged in the exploration of center vs. periphery of the arena.
- All the previously acquired behavioral tracks were reanalyzed to assess whether any difference in the center vs. periphery exploration was present between genotypes. This novel analysis allowed us to exclude the hypothesis that a combination of

abnormal anxiety and activity levels might be related to altered visual depth perception in *miR-132/212* null mice.

- New data on spine density and neuronal soma size have been added confirming the reduction in spine density and demonstrating that this reduction is not associated with changes in pyramidal neuron soma size.

Critically, the differences reported between genotypes are of smaller magnitude than the pervasive and significant discrepancies with these same techniques presented (and reproduced) relative to the relevant literature on both binocular matching and stereopsis. Overall, this manuscript fails to reproduce essential changes in binocular matching reported by several groups in multiple papers that established this line of inquiry. Thus, most of the conclusions presented in the manuscript are not supported by good evidence.

A: The reviewer makes two important points about the comparison between our ephys and behavioral data and those present in the literature.

Concerning binocular matching data, we would like to point out that our age-range (P27-28) was more precocious than that present in some of the published papers. Specifically, the closest age range in the literature is the P26-P27 group in Wang et al., 2013 that shows in Table S1 a mean ΔO value of 28.3 ± 2.5 deg that is not different from our P27-28 data. To further investigate this point, we tested whether a residual development of binocular matching could occur after P27-28 as suggested by the reviewer (see next point). To address this issue we performed new experiments assessing ΔO in fully developed P60 wt mice. The results showed a mean ΔO of 21.8 ± 2.4 deg in agreement with the published data for mature mice. This value is dramatically different from the ΔO observed in P60 *miR-132* mutant mice, underscoring the impaired binocular matching present in these mutants. Overall, the new data and the convergence of the new subject-based analysis of binocular matching (Fig. S2B) with the previous unit-based analysis provide very robust evidence that binocular matching is disrupted in *miR-132/212* null mice. Furthermore, the assessment of binocular matching at two different ages makes the manuscript much stronger because 1) it allows to conclude that the binocular matching impairment in *miR-132* KO mice is already apparent at initial stages of binocular matching maturation and persists in adult KO mice; and 2) it allows to compare ΔO of mice with normal visual stimulation with age-matched monocularly deprived mice. The results of this comparison showed that the effects of monocular deprivation on binocular matching is occluded by loss of *miR-132*, suggesting that *miR-132* is a key mediator of the effects of visual experience on development of binocularity.

Concerning the comparison between our behavioral visual cliff data and the data reported in the original 1965 Fox paper, there are substantial technical differences that in our opinion make the two approaches difficult to compare. In the visual cliff test reported by Fox, indeed, the author used a forced two-choice task by placing mice on a centre ridge in the visual cliff apparatus and recording the side on which mice stepped. By contrast, we measured the animals' spontaneous preference for the exploration of the deep or the shallow side of the arena. These very different approaches lead to discrete data in the descent two-choice test, and to continuous data in our test. Importantly, our time measures include periods during

which the animals are not forced to make a choice. For these reasons we believe that a direct quantitative comparison of our results with those presented by Fox is difficult to make. Importantly though, qualitative comparison between our data and Fox data shows that the results of the approaches are overlapping: the performance of animals with monocular vision was significantly worse than that of binocular animals, and standard-reared animals subjected to monocular deprivation during the critical period undergo a marked deterioration of depth perception (Blake and Hirsch, 1975; Kaye et al., 1981; Timney, 1983; Baroncelli et al., 2013; Fig. 5b and 5e of present work). Moreover, our novel experiments demonstrate that eliminating the visual depth cue eliminates any bias in the exploration of the two sides of the apparatus.

We tried to replicate exactly the same technique used by Fox in mice and rats, but we observed that animals are not really motivated to step down from the ridge unless the latter is very uncomfortable, with the risk of confounding effects derived from stressing conditions on their behavior. Accordingly, other papers used the Fox descent test, but were forced to limit to 1 the number of trials administered to each animal (e.g., Walk et al., 1957; Bauer, 1973), an approach that is of obvious limitations for analysis in single or double transgenic mice.

There are numerous mis-statements in the manuscript and incorrect referencing.

For example:

'when mouse visual function reaches adult levels of performance (P28)' (Cancedda et al., 2004; Hoy and Niell, 2015; Kang et al., 2013; Ko et al., 2013; Rochefort et al., 2011; Wang et al., 2010)

A: The referee makes a good point. We have rewritten the sentence according to the referee considerations. The new sentence reads: "Thus, we analyzed miRNAs regulated during visual cortical development by performing an RNA sequencing analysis of miRNAs expressed in the visual cortex immediately before eye opening (P10) and when molecular mechanisms underlying development of mouse visual function are fully engaged (P28)".

Figure 1.

A. Given that the authors previously published a study in Nature Neuroscience reporting that it was the decrease in miR-132 expression that is required for ODP, it is confusing that here the authors chose to compare transcript expression at P10 vs. P28, and P28 KO vs. WT. What is the reader meant to deduce from this list of pathways spanning axon guidance, synapse formation, and activity-dependent formation? These pathways are established contributors to experience-dependent cortical plasticity. This figure is simply not informative.

A: To answer this point we better explained in the revision the meaning of the KEGG analysis reported in Fig. 1. We agree that the crucial information deriving from the sequencing analysis is the list of the regulated genes, and indeed these gene lists are reported as supplementary tables S1-S3. However, a way to summarize these data using a functional classification is to group the regulated genes belonging to the same biological process, and check whether a given process is significantly enriched with regulated genes. On one hand, this procedure provides an immediate tool to understand the functional effects of the regulated genes, and on the other hand the fact that the pathways shown in fig.1B,C are established contributors to experience-dependent cortical plasticity ensures that our

sequencing and bioinformatic analyses are picking the correct genes that were expected to be regulated in this time window. For this reason we specified (p. 6) that “Many pathways previously involved in cortical development and plasticity were affected (Baroncelli et al., 2016; Berardi et al., 2000; Buffington et al., 2014; Hensch, 2005; Kobayashi et al., 2015; Levelt and Hübener, 2012; Tropea et al., 2006) strengthening the predictive validity of our analysis.”

B. The authors should list the 39 genes at the center of the Venn diagram, the magnitude of the effect and confirm these findings by immunoblot for at least the top 10.

A: We thank the reviewer for making these important points. To sort the 39 genes, we used the Fisher's combined probability, an index used to combine the p-values from several independent tests. Namely, the p-values of (1) the age dependent downregulation, (2) the upregulation in P28 mutants, and (3) the probability of being a miR-132-3p target (1-P_{CT} in Mouse Target Scan 7.1) were used. The sorted list is provided in Table S4.

The analysis of the top 10 genes by immunoblot requires validated antibodies available to the public that should be tested on knockout tissue to be fully reliable. Moreover, the gene annotation used for the sequencing does not distinguish different isoforms that could be present for differential splicing or posttranslational modifications. For these reasons, confirmation of RNA seq data is not normally performed by immunoblot. On the other hand, analyzing molecular changes occurring in the cortex at protein level is indeed very important. For this reason, we have started a project to analyze the effects of our manipulation at proteomic level, however we think that this issue will be better analyzed in a separate future manuscript.

Figures 2

A. The OSI (.6 here vs. ~.9), DSI (.35 here vs. ~.2) and OS tuning width (35 here vs. ~22) differ substantially and significantly from the adult values reported elsewhere, including a detailed study by the Niell lab, Hoy et al. 2015. Given that the authors state in the methods that they are using the same approach as the Neill and Stryker, 2008, this discrepancy must be resolved. The likely reason is that the authors are not examining mice with mature physiologic properties despite their assertion (l. 86). This choice of experimental design attenuates interpretation of the experiments presented in the manuscript.

... and

Figure 3

A. The deltaO presented here for WT mice conflicts with published reports. These discrepancies are not mentioned or explored. Here the values for WT mice are reported as ~30 degrees. Wang et al, Neuron 2010 report values closer to 17 degrees. This Neuron paper established this line of inquiry. The values reported here are beyond the range for the normal developmental trajectory but similar to those for dark-reared mice. Given these problems, it is unclear that the magnitude of difference reported by the authors is stochastic variation because the authors fail to report the number of mice examined. For comparison, Wang et al. examined 15-18 mice per group.

A: In these two points, the reviewer asks for a comparison between the absolute values of OSI, tuning width, DSI, and ΔO that we measured, and the values present in the literature. Concerning the comment on the number of animals, we now report the numbers together with number of cells in figure legends.

OSI, tuning width and DSI: it is important to note that, as we stated in the first submission, our multielectrodes are different from those used by Niell and Stryker 2008. Indeed, they are made of two shafts containing two tetrodes inserted at depths sampling from layer III to layer V of binocular visual cortex. Thus, our OSI measurement (Fig. 2C) of 0.69 ± 0.02 receives a substantial contribution from layer V units. This is in line with the results by Niell and Stryker 2008 that showed that OSI in layer V is 0.6 whereas it is 0.82 in layer II-III, and in 0.83 in layer IV. Similarly, tuning width was also lower in layer V (Niell and Stryker, 2008, tuning width 38 ± 5 deg) than in other layers. Average DSI value in the different layers was not explicitly reported in Niell and Stryker 2008, but Fig. 5C clearly supports the statement made by the authors that “almost all direction-selective units were putative excitatory units in layers 2/3 and 4.”

It is also worth noting that other groups measured OSI in conjunction with binocular matching with results comparable to ours: Wang et al by Cang's group in 2010 found an OSI of 0.69 ± 0.03 at P31-P36 (a value exactly overlapping with our OSI), and 0.76 ± 0.03 at P60-P90, but in their more recent paper published in Neuron they found values comprised between 0.61 and 0.68 at all ages tested (from P20 to P90, see Supplementary table 1 of Wang et al. 2010); and in the PNAS 2015 paper by Krishnan et al. OSI values for wild-type mice were reported to be around 0.5 (see Fig. 7E). It is likely that these differences in the absolute values of these parameters are due to different sampling across the cortex or possibly to differences in spike sorting and electrodes used. We have specified in the Methods that “The recording sites were located between layer III and layer V, therefore the physiological data recorded mostly reflects the properties of these layers”.

Binocular matching: Concerning ΔO , our data obtained in P27-28 mice were comparable to the P26-P27 group in Wang et al., 2013 (Table S1) that shows a mean ΔO value of 28.3 ± 2.5 deg. The ΔO of the more mature age groups analyzed in Wang et al., 2010 (P31-36 and P60-90) was decreased raising the possibility of further development of binocular matching after P30. Following the reviewer's suggestion, we tested whether a residual development of binocular matching could occur after P27-28 in our mice assessing ΔO in fully developed P60 wt mice. The results showed a mean ΔO of 21.8 ± 2.4 deg in agreement with the published data. This value is dramatically different from the ΔO observed in P60 miR-132 KO mice, strengthening the notion that the lack of miR-132 leads to an impaired maturation of binocular matching. The new data are reported in Fig. 3A and S2B.

B. The impaired binocular matching is a fairly interesting result but there are extensive concerns that this finding may not be specific for miR-132, but a general consequence of transcriptional mis-regulation. This possibility is not raised or addressed in the manuscript. This should be addressed experimentally.

A: We thank the reviewer for raising this point. To investigate the possibility of a global transcriptional mis-regulation we measured soma size of visual cortical pyramidal neurons. Indeed, the presence of a generalized transcriptional misregulation is usually reflected by

changes in cell size (Li et al., 2013; Lin et al., 2012 and Lovén et al., 2012). We did not find any change in the soma size of miR-132 KO layer V pyramidal neurons (wt $121.6 \pm 3.2 \mu\text{m}^2$ N = 3 mice, n = 92 cells; miR-132 KO $123.5 \pm 3.3 \mu\text{m}^2$ N = 3 mice, n = 90 cells; t-test p = 0.68) in agreement with previous data showing that miR-132 antagonization in pyramidal neurons resulted in reduced spine density, despite normal neuronal size and arborization (Mellios et al., 2011). We also think that global misregulation is unlikely because bioinformatic analysis showed that the genes that are up-regulated in the miR-132 KO are significantly enriched in miR-132-3p predicted targets (odds ratio 5.07; Fisher exact test p < 0.0001, Table S3) suggesting a specific effect of miR-132 deletion on its targets.

Minor:

i. the number of units and mice examined should be included in the figure legend as well as the statistical test employed.

A: We have added the number of units and mice in the legends.

Figure 4

A. Panels A – C. The finding that neutralizing miR-132 impairs ODP was published in 2011 with calcium imaging in vivo (Mellios et al., 2011). That the authors can reproduce this finding by genetic deletion of miR-132 is an incremental addition.

A: We thank the referee for giving us the opportunity to better clarify the differences between our work and the Mellios et al paper. Mellios et al obtained this fundamental data using viral transduction of a miR-132 sponge, while we employ a genetic approach. The results of Fig. 4A-C are in our opinion crucial controls that ocular dominance plasticity is also blocked using our approach. Only having established this point, we could use our genetic model to test the main question of our paper, i.e. investigating the impact of mutating a key plasticity molecule on visual cortical development. We have rewritten the section of the Results titled “MD does not affect OD and binocular matching of orientation preference in miR-132/212 null mice” to make this point clear.

B. Panel B. The authors should present the ODP as contralateral bias index per mouse, not ODI values combined for all units across all mice. CBI is the standard in the field, and has been employed by the Pizzorusso lab several publications.

A: We followed the reviewer’s indication showing the CBI for each animal in Supplementary figure 3A. The novel statistical subject-based analysis completely reproduced the results of single-unit data.

C. Panel D. Again, the values for deltaO reported here differ significantly from other studies. Moreover, the magnitude of the effect is far less here. Wang et al report that MD prevent the decrease in deltaO from 30 degrees to ~17 degrees, nearly a halving of deltaO. That is twice the magnitude of effect reported here. Whether these experiments have been performed and/or analyzed properly is a pervasive concern.

A: Concerning the comparison between our ΔO values in those published in the literature for fully developed mice please see answer to Fig. 2 and Fig. 3 points. We have now added the P60 wt data in Fig. 3A that are in line with those present in the literature for mature mice. We have also added the animal-based statistical analysis of ΔO in Suppl. Fig. 3B. The results

confirm that the impairment present in non-deprived miR-132 null mice is not different from that present in MD wt mice, and it is not further enhanced by MD.

Minor:

i. 'Single-unit analysis of the ocular dominance index (ODI; 259 (Niell and Stryker, 2008) confirmed the lack of OD plasticity in null mice'

- This reference is incorrect. Niell and Stryker do not examine ODP or introduce ODIs. ODIs are typically presented for experiments employing intrinsic imaging, such as Cang et al. 2005.

A: We thank the referee for noticing this mistake. We eliminated the incorrect quotation.

Figure 5.

A. This assay seems unlikely to report depth perception. This original work on depth perception in mice (Fox, 1965) revealed that mice prefer the shallow side to the deep side 90% of the time (10 mice). That corresponds to a DI, the metric presented here, of 0.8, TWICE that reported by the authors. Converting the DI of 0.4 to percent preference corresponds to a preference of the shallow side in this study of 70%. Chance distribution of the two sides is 50%, yet the p value is < .001 for ten mice by the analysis presented by the authors. This seems dubious.

A: The visual cliff test used in this work takes advantage of a totally different approach with respect to that reported in the Fox paper mentioned by the reviewer, making the two measurements difficult to compare at quantitative level. In the visual cliff test reported by Fox, the author used a forced two-choice task by placing mice on a centre ridge in the visual cliff apparatus and recording the side on which mice stepped. Each mouse performed ten trials. In contrast, we evaluated the animals' spontaneous preference for the exploration of the deep or the shallow side of the arena measuring the time spent on each side. These very different approaches lead to discrete data in the Fox descent two-choice test, and to continuous data in our test. Importantly, our time measures include periods during which the animals are not engaged in choosing between the two sides. For these reasons we believe that a direct quantitative comparison of our results with those presented by Fox is difficult to make. However, qualitative comparison between our data and Fox data shows that both approaches require binocular vision: the performance of animals with monocular vision was significantly worse than the performance of binocular animals; and standard-reared animals subjected to monocular deprivation during the critical period undergo a marked deterioration of depth perception (Blake and Hirsch, 1975; Kaye et al., 1981; Timney, 1983; Baroncelli et al., 2013; Fig. 5b and 5e of present work). To further validate our test, we assessed the behavior of binocular wild-type mice (N = 7) in the visual cliff apparatus in the absence of visual cues, with visual stimuli placed immediately below the glass plates. In this condition, binocular wild-type animals equally explored the two sides of the arena and a significant difference was detected with respect to the visual cliff condition (t-test, $p < 0.01$; Fig. 5C). This new experiment demonstrates that removing the visual depth cue eliminates any bias in the exploration of the two sides of the apparatus.

We tried to replicate exactly the same technique by Fox in mice and rats, but we observed that animals are not really motivated to step down from the platform unless the latter is very uncomfortable, with the risk of confounding effects derived from stressing conditions on their

behavior. Accordingly, other papers used the Fox descent test, but they were forced to limit to 1 the number of trials administered to each animal (e.g., Walk et al., 1957; Bauer, 1973), an approach that is of obvious limitations for analysis in single or double transgenic mice.

Finally, the p-value of our statistical analysis of the comparison between wt and ko animals is < 0.05 (not < 0.001) both at P30 and at P60 (see fig. 5E, H).

- The data should be presented and the average percent time each mouse dwells on the shallow side. A data point should be presented for each mouse. The stats should be recalculated.

A: As suggested by the reviewer, we show the average % time spent exploring the shallow side for each mouse (Suppl fig. 5C,E). The statistics on these data confirm the previous analyses.

B. The principal findings of this paper are that WT mice exhibit a preference of the shallow side of 70% while KO mice has a lower preference of only 60% (DI = 0.2). To be frank, this 10% difference is not compelling or interesting.

A: We thank the referee for raising this potentially confounding point. This is the first time in the literature that an impairment of binocular matching of orientation preference of cortical neurons has been related to a behavioral failure of animals, indicating a physiological requirement of this neuronal feature in visual perception. We think that the novelty and the relevance of our behavioral data reside in this point.

The effect we report may seem small at a first glance however two considerations should make clear that it is both significant and considerable. (1) The first consideration is that our scale intrinsically compresses the changes into a small range. The behavioural measure can only be attributed to two categories (time spent in the shallow side and time spent in the deep side) implying that all possible outcomes (from depth-blindness to full ideal observer behaviour) range only between 50% and 100%. The fact that we employ a less stressful naturalistic and spontaneous exploration task further diminishes the possible range of outcomes as the mouse bases his exploration on several factors, and it is nearly impossible that even a mouse with fully developed depth perception spends all of his time in the less shallow part of the arena. Moreover, binocular matching is likely to be only one of the functional features of cortical neurons contributing to the building of depth-perception capacities and other visual cues could provide the animals with residual depth perception of the visual cliff. (2) The second consideration results from transforming the percentages into sensitivity values. Sensitivity is a widely used measure in psychophysics and allows estimating the signal to noise ratio in a given task from the raw percentages. This analysis shows that null mice have a sensitivity of 0.25 whilst WT mice have a sensitivity of 0.5, which is a two-fold increase. So despite the change between 60 and 70% may not be stunning at a first glance it is indeed is a strong effect.

C. The DI for WT mice is 0.4 in panel E and almost 0.6 in panel H, essentially the same magnitude of difference the authors assert separates WT and KO mice. Some explanation is required.

A: We thank the referee for asking to comment about the different performance in visual cliff between wt and KO at P30 and P60 (Fig. 5E, H). The difference in exploration of the shallow side between P30 and P60 in wt (72% vs 78%) could be due to some residual maturation of depth perception, and is in line with the binocular matching maturation that we have now included in fig. 3A. Importantly, the magnitude of the difference between P30 and P60 wt mice is not comparable to the difference between wt and ko animals both at P30 and even more at P60: indeed the percentage of time spent exploring the shallow side is 72% for wt and 61% for KO at P30; and 78% for wt and 64% for KO at P60.

D. The number of approaches to the center point differs dramatically between genotypes. This assay is likely reflecting a combination of anxiety, exploration and other cues. In addition, given that the mice can palpate the glass surface with their whiskers, a definitive perception, it is questionable that this assay reports depth perception.

A: We thank the referee for prompting us to analyze in depth the anxiety and exploratory activity in our visual cliff test. To assess this issue we (1) analyzed the general motor activity and anxiety-related behavior of wt (N = 7) and null mice (N = 9) in the visual cliff apparatus in the absence of visual depth cues, i.e. with visual stimuli placed immediately below the glass plates. The time spent by mutant mice in both the central and peripheral portion of the apparatus was not different from that recorded for wt animals (t-test, $p = 0.663$; Fig. 5D). (2) All the previously acquired behavioral tracks were reanalyzed to assess whether any difference between genotypes in the center vs. periphery exploration was observed in the presence of the visual depth cue. No significant alteration in center vs. periphery bias was observed between the two experimental groups (t-test, $p = 0.177$), proving that no differences in anxiety or exploration activity could be responsible for the different performance of miR-132 KO and wt mice.

Concerning the possible influence of nonvisual cues, such as tactile perception, the glass floor is the same for the two sides excluding that tactile texture might differentiate the shallow side from the deep side of the arena. Moreover, all non-visual cues do not contribute to the preferential exploration of the shallow side because in the experiment performed with visual stimuli placed at the same height immediately below the glass plates, no preferential exploration was observed (Fig. 5C) in wt and KO mice. These data strengthen the importance of visual cues for this test that was also indicated by the loss of preferential exploration of the shallow side observed when one eye has been shut (mon and wt-md groups in fig. 5B,E), despite that all other sensory cues remain unaltered.

Figure 6.

A. Are these neurons in V1? What is the evidence these neurons are in V1? It is widely appreciated that EGFP-M does not express GFP in V1 (Holtmaat, 2005).

Hofer et al, Nature 2009 screened scores of mice to find a handful that expressed a few neurons in V1. How have the authors made this determination? Some verification must be presented

We thank the reviewer for prompting us to better clarify our spine measurement protocol: We explain in the Methods that “A low magnification image was first acquired (an example is now reported as Suppl. Fig. 6) to guide the experimenter to acquire a high magnification

image in V1 based on anatomical points of reference (see sketch of a cortical slice from mouse brain atlas at the same A-P level, image included in Suppl. Fig. 6).” As shown in the low-magnification figure there are scattered cells in V1, in agreement with the paper by Hofer et al who used this mouse model to analyze spine dynamics in V1. The number of mice we employed (now 13 per experimental group) is similar to that used by Hofer et al to reveal MD effect (N = 7-13 see Fig.1 legend for Hofer et al.).

B. Panel A. This image quality is substandard. In addition, it does not represent the spine density reported in panel B. How this image has been processed must be detailed in the methods section, particularly any non-linear modifications.

We have added a new image that better represents the data. We specify that the only manipulation that was applied to the figure was the linear stretching of image contrast.

C. Panel B reports more than 1.5 spines per micron for L5 neurons. This value is 3X that reported elsewhere including Hofer, 2009 (see Fig. 3a). This spine density is dramatically higher than the bulk of the available literature and raises substantive concerns about the accuracy of this analysis.

We thank the reviewer for drawing our attention to this important point. The reviewer is absolutely right that our values of spine density were beyond the variability that is present in the literature. For this reason we fully reviewed our ImageJ macro and indeed we found that the pixel-to-micron calibration value that was loaded was related to an imaging condition with a different zoom factor. We reanalyzed all the images with the correct zoom factor and the new results are reported in Fig. 6A. The new values are in line with those described for layer 5 apical dendrite of pyramidal neurons in the primary visual cortex (e.g. Kim et al., 2016). The reduction present in miR-132 KO mice is still the same and statistically significant.

D. Panel B, the magnitude in the difference in spine density is approximately 10%. This is not convincing, particularly given the small sample examined and the quality of the representative images.

The spine density reduction that we observed in mutant mice (a 14% decrease) is consistent with previous studies showing that mir-132 levels are closely related to spine density in different brain regions (Hansen et al., 2010; Mellios et al., 2011; Tognini et al., 2011; Pathania et al., 2012). To further investigate the significance of the effect on dendritic spines we increased the number of animals analyzed. Even with a larger sample, the effect was still significant. It is worth noting that effects of similar size have been shown in models of brain disease, e.g in layer V neurons of visual cortex of models of Angelman Syndrome (Kim et al., 2016) and dentate gyrus neurons of Down syndrome model (Catuara-Solarz et al., 2016).

E. Panels C-D suffer from many of the same shortcomings as Figures 2 and 3. In addition, the authors must provide a demonstration that they can detect recombination of the miR-132 locus because EMX1:Cre displays germline recombination at high frequency.

A: We now show the ΔO values for single mice (Fig. S7) and we included the statistical comparisons across the averages between mice. This analysis confirms the results of our previous statistical analysis based on individual units. Please refer to the answer to fig. 2 and

3 comments for a discussion of the comparison of our data with the absolute values present in the literature.

We exclude the presence of germline recombination in Emx1:Cre mice because:

1. miR132/212^{+fl} male mice were crossed with Emx1:Cre females to generate a mouse line carrying the floxed *miR132/212* and Emx1:Cre alleles; we used this specific breeding strategy to avoid that undesired germline recombination could occur in the Emx1-Cre testis, a tissue known to express Emx1 (Iwasato et al., 2004). We included information about breeding strategy in the description of experimental procedures (see supplemental information).

2. routine tail genotyping of these mice was carried out using primers able to distinguish the floxed allele from the wild type and the deleted *miR132/212* alleles (Remenyi et al., 2013, Fig. 1). Indeed, the 3 primers used result in bands of 373 bp for the wild-type allele, 420 bp for the floxed allele and 550 bp for the deleted allele. Only mice in which germline recombination could be excluded because of the presence of the 420 bp floxed band in the tail tissue were analyzed. This information has been now included in experimental procedures (supplemental information).

REVIEWER 3

It was a pleasure to read the manuscript by Mazziotti et al. The manuscripts detail the investigation of small RNA expression and posttranscriptional regulation in the visual cortex of the developing mouse. The authors identified miR-132/212 as being both altered and influential on the transcriptome among other miRNA, many of which have been reported previously in other brain regions. This miRNA family is also known for its activity-associated expression pattern and was clearly ective the authors used a combination of global and tissue specific knockout model an important target for further functional characterisation. To achieve this obj's to explore the functional significance in visual processing. This part of the manuscript is outstanding.

A: We greatly appreciate the reviewer's interest in our manuscript.

The authors were quick to focus on miR-132/212, but I was left wondering about the influence of other developmentally regulated miRNA? These miRNAs were way down the list of developmentally altered miRNA both by fold change and p-value, with miR-29 and miR-219 for example, showing much greater change. I could not see the qPCR validation of miR-132 or 212 but this data was available for miRs 29 and 219? Incidentally, the corrected p-values (or FDR) should also be provided for each of the tabulated miRNA and transcripts. Perhaps more detail on the rationale for this selection could also be provided. One wonders if this was just a convenient introduction for established animal models and experiments?

A: We agree with the referee that other miRNAs could play an important role in cortical development. The cortical tissue is composed of many different cell types (astrocytes, microglia, endothelial cells, different types of neurons) and it is possible that different miRNAs can play a regulatory role in a cell- and age-specific manner. Grouping together the different isomiRs in the miRNA precursor's list, miR-132 was preceded, (using ordering by p-values), by: (1) miR-134 that however had a much smaller fold-change; (2) by miR-298 for which no information about its role in the brain development is available; (3) miR-219 and miR-338 which have been suggested to be key players in brain myelination (Dugas et al., 2010; Zhao et al., 2010), a process that is very active during the exact developmental time window that we analyzed; and (4) miR-29a, a miRNA that is present in both excitatory and inhibitory neurons and which has been involved in neurodegeneration and aging. However, we thought that miR-132, with its specific expression in excitatory cells, its well established regulation by visual experience, and the availability of conditional KO mice, was an optimal first candidate to reveal neuronal-subtype specific roles of miRNAs in visual cortical development. Nevertheless, we are keenly aware of the potential importance of the other miRNAs for cortical development; indeed we are currently investigating the role of miR-29 and miR-219/miR-338 in neuronal and oligodendrocytic development in a different project. As requested we have introduced in Suppl. Fig. 1 the validation for miR-132.

To correct for multiple comparisons and the multiple hypothesis testing problem, we added the posterior probability of differential expression (PPDE) that is computed from the distribution of the raw p-values, by fitting a mixture of beta distributions to this distribution, and then computing the probability that the observed p-value belongs to the component in

the mixture associated with differentially expressed genes [P. Baldi and G. Wesley Hatfield. DNA Microarrays and Gene Regulation—From Experiments to Data Analysis and Modeling. Cambridge University Press, (2002)]. Genes of interest having a PPDE above 0.6 are considered significant. We adopted this test because of the well-known overcorrection associated with the alternative Bonferroni correction. **Most importantly, we would like to stress that our selection of the candidate genes mediating development of binocular matching does not result from a single comparison or p-value result, but rather it relies on the convergence of multiple lines of evidence, including: being regulated during development, regulated in the miR-132 KO, and predicted to be a direct miR-132 target.**

I also wonder how many target genes altered during development of the visual cortex were influenced by multiple miRNA. Would it be possible for the authors to map the network architecture of posttranscriptional interaction during development of the visual cortex? How were miRNA targets predicted? It would be nice if the quality or strength of the putative interaction were tabulated. Are these conserved targets or have non-conserved targets been included?

A: We very much appreciate this comment. To show the action of different miRNAs on the same target, we have included in supplementary table S2 a list the developmentally regulated genes targeted by more than two miRNAs and displaying the identity of these miRNAs.

We specified in the methods that target prediction was performed using existing database scoring the strength of the interaction (Target Scan Mouse 7.1). Genes with only poorly conserved sites were not included. Similar data were obtained using an alternative method to identify targets combining three different sources:

1) TargetScan:

The “Broadly conserved” category of datasets corresponding to conservation among vertebrates was used. The conservation score used by TargetScan is PCT defined in: Friedman, Robin C., et al. Most mammalian mRNAs are conserved targets of microRNAs. *Genome Research*, 19.1 (2009): 92-105.

2) microRNA.org:

Targets were identified from target sites found by MiRanda with a threshold miRSVR score (≤ -1.2). It is possible to identify non-conserved sites in this database. However, since we take the intersection of sources, the site was found to be conserved in either TargetScan or using BBLs. MirSVR score is defined in: Betel, Doron, et al. Comprehensive modeling of microRNA targets predicts functional non-conserved and non-canonical sites. *Genome Biology*, 11.8 (2010): R90.

3) miRanda and BBLs:

Targets are identified from target sites which were found by MiRanda and filtered on a BBLs ≤ 1.0 from mm10 multiz60way. The BBLs score is defined in: X. Xie, P. Rigor, and P. Baldi. MotifMap: a human genome-wide map of candidate regulatory motif sites. *Bioinformatics*, 25, 2, 167-174, (2009).

I would like to have seen more cross-referencing to previous studies of mammalian cortical development. These miRNA all seem familiar to me but it would be nice to

see their correlates in other cortical structures. Are there any specific to the visual cortex or is there a difference in the levels or timing? Have any of these been associated with neuropathology?

A: The reviewer is asking good questions. Mir-132 has been studied in the context of development of the visual cortex, but our previous in situ data showed that its expression at P28 is present also in non-visual cortical areas (Tognini et al., 2011). However, its function in these non-visual areas of the brain is still unknown. Other papers that are now cited in our manuscript have investigated its role in learning and memory in the adult. Furthermore, miR-132/212 dysregulation has been associated with a number of neurodegenerative disorders, including Alzheimer's disease and Huntington's disease, and neurocognitive disorders, including autism, Rett syndrome, and schizophrenia. This is now addressed in the discussion in the revised version in the Discussion.

It would be nice to have a bit more explanation of terms used in the figure legends. As an outsider to the field I had to refer back to the text to decipher the results in each panel.

A: We have expanded the figure legends to clarify some of the key terms.

The authors speculate on the target genes driving the miR-132 /212 associated changes in visual function. Would it be possible to recapitulate or rescue the miRNA-associated phenotype in vivo, by directly modulating some of these target genes?

A: We agree that this is a natural follow-up step for our work. However, the time required for the analysis of a single gene resulting from our study would be more than one year, considering that: (1) the tools for the manipulation of the selected genes (testable mutant mice, viral vectors) are not readily available; (2) even once available, these tools require extensive validation; and (3) additional in vivo electrophysiological experiments require considerable time. For all these reasons, we think that these studies should be carried in a separate dedicated project.

REVIEWERS' COMMENTS:

Reviewer #1 (Remarks to the Author):

The authors have fully addressed all my comments. The new data added to the revised version have improved the manuscript further.

Reviewer #3 (Remarks to the Author):

I am satisfied with the revised manuscript.